# LISA: a lightweight stratospheric air sampler

Joram J.D. Hooghiem[1], Marcel de Vries[1], Henk A. Been[1], Pauli Heikkinen[2], Rigel Kivi[2], and Huilin Chen[1]

[1]Center for Isotope Research (CIO), Energy and Sustainability Institute Groningen (ESRIG), University of Groningen, Nijenborgh 6, 9747 AG Groningen, The Netherlands.
[2]Finnish Meteorological Institute (FMI), Earth Observation Research, Tähteläntie 62 99600 Sodankylä, Finland.

**Correspondence:** Huilin Chen (huilin.chen@rug.nl)

**Abstract.** We developed a new LIghtweight Stratospheric Air sampler (LISA). The LISA sampler is designed to collect four bag samples in the stratosphere during a balloon flight for $CO_2$, $CH_4$ and CO mole fraction measurements. It consists of 4 Multi-Layer Foil (MLF) sampling bags, a custom-made manifold, and a diaphragm pump, with a total weight of $\sim 2.5$ kg.

A series of laboratory storage tests were performed to assess the stability of $CO_2$, $CH_4$ and CO mole fractions in both MLF and Tedlar bags. The MLF bag was chosen due to its better overall performance than the Tedlar bag for the three species $CO_2$, $CH_4$ and CO. Furthermore, we evaluated the performance of the pump under low pressure conditions to optimise a trade-off between the vertical resolution and the sample size.

The LISA sampler was flown on the same balloon flight with an AirCore in Sodankylä, Finland (67.368°N, 26.633°E, 179 m.a.s.l.) on 26 April and 4 – 7 September, 2017. A total of 15 stratospheric air samples were obtained during the ascent of four flights. The sample size ranges between 800 to 180 mL for the altitude between 12 and 25 km, with the corresponding vertical resolution ranging from 0.5 to 1.5 km. The collected air samples were analysed for $CO_2$, $CH_4$ and CO mole fractions, and evaluated against AirCore retrieved profiles, showing mean differences of 0.84 ppm for $CO_2$, 1.8 ppb for $CH_4$ and 6.3 ppb for CO, respectively.

High-accuracy stratospheric measurements of greenhouse gas mole fractions are useful to validate remote sensing measurements from ground and from space, which has been performed primarily by comparison with collocated aircraft measurements (0.15 – 13 km), and more recently with AirCore observations (0 – 30 km). While AirCore is capable of achieving high-accuracy greenhouse gas mole fraction measurements, it is challenging to obtain accurate altitude registration for AirCore measurements. The LISA sampler provides a viable low-cost tool for retrieving stratospheric air samples for greenhouse gas measurements that is complementary to AirCore. Furthermore, the LISA sampler is advantageous in both the vertical resolution and the sample size to perform routine stratospheric measurements of the isotopic composition of trace gases.

# 1 Introduction

The stratosphere plays an important role in the climate of the earth and is affected by ongoing climate change. Changes in stratospheric ozone and water vapour levels in turn affect climate and climate variability (Baldwin et al., 2007). The distribution of trace gases in the atmosphere provides useful insights in atmospheric transport and chemistry. Stratospheric tracer observations are essential for validation of General Circulation Models (GCMs). The stratospheric meridional overturning, or the Brewer–Dobson circulation (BDC), was predicted to increase in strength from modelling studies (Butchart, 2014). The mean age of stratospheric air samples was shown to be a good diagnostic for the strength of the BDC; however, no significant change in the strength of the BDC in the northern hemisphere at mid latitudes was detected (Engel et al., 2009, 2017). In spite of all the efforts to make observations of stratospheric tracers, GCMs remain poorly constrained (Gerber et al., 2012), a problem already pointed out several decades ago (Ehhalt et al., 1983).

In order to determine the vertical distribution of trace gases, both remote sensing techniques and airborne platforms are utilised. Remote sensing is performed either from the ground, e.g. TCCON (Wunch et al., 2011), and from satellite instruments like SCIAMACHY (Frankenberg et al., 2011) and IASI (Crevoisier et al., 2013). Although remote sensing techniques have a high spatial and temporal coverage, they are subjected to systematic bias and need to be calibrated. Calibration requires in situ measurements, of which the availability relies on infrequent campaigns (e.g., Engel et al., 2016).

In situ measurements of stratospheric air up to 35 km above mean sea level (a.m.s.l.) can currently only be achieved on balloon-borne platforms. To this end, both airborne analysers (e.g., Daube et al., 2002) as well as sampling techniques have been developed specifically for balloon-borne platforms. Of these techniques, the cryogenic sampling method (Lueb et al., 1975) is the most employed technique. It has been used for the analysis of many trace gases like $SF_6$, $CO_2$, $CH_4$, $N_2O$ and halocarbons (Aoki et al., 2003; Engel et al., 2002; Laube et al., 2010; Nakazawa et al., 1995, 2002) and isotopic composition measurements (Kaiser et al., 2006; Rice et al., 2003; Röckmann et al., 2011; Sugawara et al., 1997). As outlined in Fabian (1981), cryogenic sampling overcomes the problem of small samples that are obtained from the grab sampling technique. Typically, cryogenic samplers retrieve ~15 samples of 10 to 20 litre at STP (Standard Temperature and Pressure) (Fabian, 1981; Honda, 2004; Lueb et al., 1975) with a sufficiently good vertical resolution of ~1 km (Schmidt et al., 1987). This makes the cryogenic sampling technique suitable for multi-tracer and isotopic composition analysis. Secondly, the cryogenic technique provides a way of contamination-free sampling.

Inasmuch as their accuracy, these samplers and airborne analysers are heavy weight (100 – 250 kg), which requires sophisticated planning and significant resources for a single launch. As a result of the intensive operation, stratospheric observations have been sparsely made both in time and space. Existing sampling is mainly restricted to the Northern Hemisphere mid-latitudes and polar regions, with the tropics under-sampled. Recently, a light cryogenic sampler (22 kg) using liquid neon (Morimoto et al., 2009) was developed, and launched from a research vessel to retrieve stratospheric air samples in the tropics (Fuke et al., 2014); however, it is capable of retrieving only one sample per flight.

Recently, AirCore has been shown to be a viable method for profile measurements of greenhouse gases (Engel et al., 2017; Karion et al., 2010; Membrive et al., 2017). AirCore is much lighter (2 – 9 kg) compared to the cryogenic sampler and can

be launched on weather balloons. The launch of AirCore is also much simpler than the operation of large-payload cryogenic samplers. Being a passive sampling technique, AirCore does not provide large sample amount from the stratosphere. The volume of air sampled between 0 to 200 hPa (12 to 30 km) by the AirCore ranges from 300 to 600 mL, depending on the geometry of the AirCore. This is problematic for accurate analysis of isotopic compositions or multiple tracers. Sub-sampling of the stratospheric part of the AirCore samples has been used for measurements of $\Delta^{17}O$ in $CO_2$ (Mrozek et al., 2016) and radiocarbon analysis (Paul et al., 2016). The samples have small sample size, which limits the analytical precision of their analyses. Besides this, the vertical resolution of the samples was low and the altitude registration of these samples was associated with significant uncertainties.

In this work, we present the deployment and field-tests of a new Lightweight Stratospheric Air sampler (LISA). With the LISA sampler, we aim to develop a sampling technique complementary to AirCore. With LISA we aim for a reasonable accuracy of GHG measurements, which does not necessarily meet the WMO recommended compatibility goals of 0.1 ppm (µmol mol$^{-1}$), 2 ppb (nmol mol$^{-1}$), 2 ppb for $CO_2$, $CH_4$ and CO, respectively, but would be sufficient, e.g. to detect the large vertical gradient of $CH_4$ in the stratosphere. Moreover, we intend to obtain significantly larger amount of air samples from the LISA sampler than from the AirCore sub-sampler.

The design of the LISA sampler is described in section 2. The accuracy of the $CO_2$, $CH_4$ and CO mole fraction measurements of the LISA samples is assessed by the sample storage test that is detailed in section 3. The vertical resolution and sample amount are both a function of sampling time and are therefore discussed together in section 4. Following the development of the sampler, we present the deployment of the sampler in the field and the comparison of the $CO_2$, $CH_4$ and CO mole fraction measurements between the LISA sampler and AirCore in section 5. Finally, we give discussion and conclusions in section 6 & section 7.

## 2 LISA sampler design and operation

We design the sampler to collect stratospheric air samples during a weather-balloon flight, where the balloon typically bursts at ∼30 km altitude. The total payload of a weather balloon typically ranges between 0.2 – 12 kg. Therefore, the sampler needs to be lightweight. To achieve this, we have used bags to contain air samples instead of glass or metal flasks that are commonly used for accurate trace gas measurements, and a diaphragm pump instead of previous cryogenic coolers to pump air. Besides these, a datalogger is used to make the system fully automatic during flight. The payload is housed in a Styrofoam package for thermal insulation and to prevent it from damage during landing. Previously the use of a gas pump and Tedlar bags have been successfully used to sample air from a UAV for methane studies (Greatwood et al., 2017).

Figure 1 shows the schematic diagram of the sampler. The system consists of a diaphragm pump (KNF, product no. NMP 850.1.2 KNDC B) and four Supel Inert Multi-Layer Foil (MLF) bags (Supelco, product no. 30227-U). The pump has a flow capacity of 8 L min$^{-1}$ at STP. The pump utilises an EPDM rubber diaphragm (35 mm diameter) and valves, and a small piece of flexible polyurethane tube. Each bag is equipped with a screw cap combo valve that requires a 180-degree turn to be opened or closed. A servomotor (Hitec, product no. HS-65HB+) operates the valve. The pump and bags are connected to a

custom-made manifold, which is made from 5 nylon Tees (Swagelok, product no. NY-400-3) and 5 union elbows (Swagelok, product no. NY-400-9), connected by Kynar tubing (Cole Palmer, product no. EW-95100-02). A fifth screw cap combo valve is placed at the outflow end as the outlet valve, allowing the manifold to be flushed prior to sampling. The pressure inside the manifold is continuously monitored by a pressure sensor (Honeywell HSCMAND015PASA5). A datalogger (Arduino Mega 2560) operates all the electronics during flight, and logs ambient atmospheric pressure and temperature data, as well as the pressure inside the manifold and temperature within the Styrofoam package. The pump requires 24 V during operation. The power is therefore supplied by eight 3 V lithium ion batteries (CR123A) connected in series. The Arduino is powered by three batteries out of the eight (9 V). The servo motors, powered with 2 separate batteries (6 V). The sampler is placed in a Styrofoam package, with a total weight of $\sim$1 kg excluding the package, and $\sim$2.5 kg including the package. The key components are summarised in Table 1.

Because the ascent rate is usually much slower than the descent rate, we take air samples during ascent to optimise the vertical resolution (see section 4). The sampling process is triggered by starting the pump when the sampler reaches preset ambient pressure levels monitored by the on-board pressure sensor. In practice, when the preset pressure value is reached during ascent, the pump is started, with the outlet valve open and the other valves upstream of the bags closed, to flush the manifold. After 20 seconds, the outlet valve is closed, and the valve upstream of one of the chosen bags is simultaneously opened to allow sampling. The sampling of air into one bag is completed when a preset maximum sampling time is exceeded or the absolute pressure in the manifold is higher than 280 hPa (see section 4), whichever comes first. The sampling process continues until all four bags are filled. At a certain altitude, the balloon will burst and the sampler falls back to the ground where the samples can be picked up and brought back to the laboratory for analysis.

## 3  Sample storage tests

The stability of trace gases in the sample container is essential for a sampler to obtain accurate measurements of the trace gases. To this end, we have investigated the stability of $CO_2$, $CH_4$, CO and $H_2O$ mole fractions of dry air samples in two types of gas sampling bags: Tedlar and Supel Inert Multi-Layer Foil (MLF). The Tedlar bag is composed of a thin polyvinylfluoride film. The MLF bag consists of several layers: polyethylene (inner layer), aluminium foil, polyethylene, aluminium (metalised) and 60-gauge nylon, which provide a moisture barrier and light protection.

### 3.1  Experiments

A total of 7 MLF and 7 Tedlar bags were prepared, with dry air (<0.003% $H_2O$) from a cylinder; the mole fractions of $CO_2$, $CH_4$ and CO are listed in Table 2. Since the mole fractions of methane are significantly lower in the stratosphere than a typical value of around 2000 ppb in the free troposphere (e.g. Rice et al., 2003; Röckmann et al., 2011), we prepared two samples (nos. 6 & 7) with low mole fractions ($\sim$120 ppm $CO_2$, $\sim$600 ppb $CH_4$ and $\sim$75 ppb CO) by diluting air from a cylinder with $N_2$. The CO mole fractions in the stratosphere are also low, but we did not make any storage test for samples with a mole fraction lower than $\sim$75 ppb.

Directly after sample preparation, the air sample is analysed for $CO_2$, $CH_4$, $CO$ and $H_2O$ mole fractions on a cavity ring-down spectrometer (CRDS, Picarro Inc., model G2401-m). During an actual balloon flight, it usually takes $3-5$ hours from sampling until the samples are retrieved and brought back to the laboratory for analysis. Therefore, we have chosen a period of 4 hours as the storage time to represent this time delay, i.e. the bags are stored under laboratory conditions ($\sim 20°C$, $\sim 1000$ hPa, ambient mole fractions of $CO_2$, $CH_4$, $CO$ and $H_2O$) for four hours before they are analysed again. The four-hour drift during storage is defined as the difference between the measurement after four hours of storage and the initial measurement: $[X]_{4hours} - [X]_{direct}$, where $[X]$ is the measured dry mole fraction.

Previous studies show that the material of Tedlar bags is prone to water vapour diffusion (Beghi and Guillot, 2006; Cariou and Guillot, 2006), which leads to humidified air samples after four hours of storage. $H_2O$ measurements are used to obtain dry mole fraction of $CO_2$, $CH_4$ and $CO$ using the water vapour corrections described in Chen et al. (2013) and Rella et al. (2013), before assessing drift of these species.

## 3.2 The storage test results

The difference between the measured mole fractions after 4 hours and those measured immediately after filling are shown in Figure 2, which captures the drift over 4 hours of storage. The drift in $CO_2$ after 4 hours is comparable for both types of sampling bags for sample nos. $1-5$, within a range of $-0.2-0.2$ ppm (Figure 2a). Low mole fractions of $CO_2$, i.e. samples nos. 6 & 7, are less stable in both types of sampling bags; however, these low mole fractions are not observed in the stratosphere, and hence the drift observed for samples nos. $1-5$ is more representative than that observed for samples nos. 6 & 7 for the storage of stratospheric air samples. The $CH_4$ mole fractions are preserved within the range of $\pm 2$ ppb for all cases for both types of sampling bags (Figure 2b). Although the Tedlar bags perform slightly better than the MLF bags, both are satisfactory for $CH_4$ measurements when considering its large gradient in the stratosphere (500-2000 ppb e.g. Röckmann et al. (2011)).

The $CO$ mole fractions appear to be stable in the MLF bags, with no clear indication of drift, independent of the mole fractions (Figure 2c). The variability of $CO$ differences may be in large part due to the repeatability of the CRDS analyser (1-$\sigma$ 7 ppb). In contrast, the $CO$ mole fractions decrease in the Tedlar bags, coupled with a significant increase of water vapour mole fractions of up to $\sim 1\%$ (Figure 2d), which is due to the high permeability of the bag material to water vapour and has been observed in previous studies (Beghi and Guillot, 2006; Cariou and Guillot, 2006). The increase of water vapour mole fractions in the MLF bags is only up to 0.01%. The observed decrease of $CO$ mole fractions in the Tedlar bags, even when its mole fractions are lower than the ambient, cannot be explained by the permeability of the bag material, as diffusion would increase the $CO$ mole fractions. Although mole fractions are corrected for water vapour to obtain dry mole fractions, we cannot exclude that there is still some remaining bias from the water vapour correction function, and this correction function was not tested before with low mole fractions. This would not affect the depicted results for the MLF bags since the water vapour content remains low. Further investigation is needed before Tedlar bags are used to collect samples for analysis of high-precision $CO$ mole fractions at the ambient level.

We found out that it is necessary to precondition the MLF bags before use, because we observed a positive offset of $\sim 12$ ppm $CO_2$, $\sim 8$ ppb $CH_4$ and $\sim 30$ ppb $CO$ between the immediately analysed results of un-preconditioned MLF bag samples

after filling and the assigned cylinder values. This contamination issue could be overcome by preconditioning the bags with $N_2$. The bags were filled with $N_2$ from a cylinder and subsequently evacuated with a vacuum pump, prior to filling with test sample. In principle ambient air can be used to flush the bags, as long as it is dry.

Based on the storage test results, we choose to use the MLF bags for our sampler. The stability of $CO_2$ and $CH_4$ mole fraction in both MLF and Tedlar bags is comparable; however, the observed CO mole fractions in Tedlar bags is less stable than those in MLF bags. In addition, the permeability of water vapour to Tedlar bags causes a significant increase of water vapour, which may affect isotopic composition, e.g. $\delta^{18}O$ in $CO_2$. Moreover, the aluminium layers of the MLF bag protect the air samples against radiation that could affect the stability of CO mole fractions. We emphasise the importance of preconditioning the MLF bags before use.

## 3.3   The uncertainty of the LISA sample measurements

We estimate the measurement uncertainty based on the laboratory storage test results and the uncertainties associated with the sample analysis. The total uncertainty of $CO_2$, $CH_4$ and CO mole fraction measurements consists of 3 terms: sampling error, drift due to storage and analysis uncertainty. There are two contributions to the analysis uncertainty: 1) analyser precision ($\sigma_i$) and 2) calibration uncertainty ($\sigma_c$).

The sampling error encompasses any contamination introduced by the sampling system itself. This includes chemical production of the species of interest and residual air in any dead volumes of the manifold. The chemical production during sampling is likely to be very small for two reasons. First of all, the wetted surfaces, Kynar and EPDM diaphragm, are chemically inert. Secondly, the high flow rate minimizes exposure of the sample to materials used in the sampler and hence chemical interaction with the wetted surfaces is limited. In addition, the flushing procedure with high flow-rates ensures multiple turnovers of the manifold, which reduces the surface effects on the sample. These effects are thus assumed to have no influence on the $CO_2$, $CH_4$ and CO mole fractions.

The dead volume in the tube connecting the bag to the manifold is a potential source of contamination bias. The dead volume is estimated to be 1.5 mL per sample and will be at local ambient pressure prior to sampling. The dead volume uncertainty, $\sigma_v$, is estimated using a dead volume of 1.5 $mL_{stp}$, which prior to sampling is assumed to be at 50 hPa and 220 K. This volume might remain unflushed. hence the air is of tropospheric origin, with concentrations of 400 ppm $CO_2$, 1800 ppb $CH_4$ and 150 ppb CO. The total sample volume is 200 $mL_{stp}$ and has mole fractions of 395 ppm $CO_2$, 500 ppb $CH_4$ and 30 ppb CO. The bias is then calculated as the resulting deviation after mixing the contamination with sample air.

The main factor likely to affect mole fraction measurements of the stratospheric air samples is the drift in the sampling bags, an effect that has been quantified in subsection 3.1 & subsection 3.2. In principle, one could correct for the drift as a systematic error. The drift, a consequence of diffusion through the bag's material, is governed by Fick's first and second law. A systematic correction for the drift would require the determination of the species-dependent diffusivity, which usually depends on pressure and temperature. Secondly, since diffusion depends on the concentration gradient, mole fractions of the sampler environment are needed. Hence, detailed information about storage conditions over the $3-5$ hour period between sampling and analysis are required to correct for the drift. Moreover, the information required is usually unavailable in the field. Therefore, we have not

determined a correction function for the drift, but rather use the maximum observed drift as an estimate of the drift uncertainty, $\sigma_d$. The maximum observed drift in these tests were 0.11 ppm, 2 ppb and 2.7 ppb for $CO_2$, $CH_4$ and CO, respectively.

Assuming Gaussian error propagation, we compute a total uncertainty on the measurements:

$$\sigma_s = \sqrt{\sigma_d^2 + \sigma_i^2 + \sigma_c^2 + \sigma_v^2} \tag{1}$$

The total uncertainty includes the analyser precision (1-$\sigma$ 0.04 ppm, 0.2 ppb and 7 ppb for $CO_2$, $CH_4$ and CO, respectively), the calibration uncertainty (0.07 ppm, 1 ppb, 2 ppb for $CO_2$, $CH_4$ and CO, respectively), and the aforementioned drift. The different uncertainties are summarised in Table 3. We compute the total uncertainty of the LISA sample measurements to be 0.14 ppm, 2.3 ppb and 7.8 ppb for $CO_2$, $CH_4$ and CO, respectively.

## 4   The vertical resolution and the pump performance

The vertical resolution of each individual air sample depends on the vertical speed of the sampler during flight, and the effective sampling time, i.e. when the flow rate into the sampling bag is positive. On the other hand, the amount of air sample collected into each sampling bag is determined by the sampling time and the sampling flow rate. Due to the trade-off between the vertical resolution and the sample size, we evaluate the pump performance to assist the choice of the sampling time.

Under laboratory conditions the KNF pump can maintain a flow rate of up to 8 $L_{stp}$ $min^{-1}$. The performance of the small diaphragm pump is to our best knowledge not previously investigated under the atmospheric conditions in the stratosphere, e.g. in low temperature and low pressure conditions.

We evaluated the sampling performance using a simplified version of the sampler under simulated conditions in the laboratory. The test version of the sampler consisted of the KNF pump, the outlet valve and one sampling bag, supported by the required electronics (pressure and temperature sensors, a datalogger and batteries). The test version was placed in a 50-litre vessel for testing. The pressure inside the vessel was regulated by a vacuum pump, mimicking the atmospheric pressure levels in the stratosphere. After a desired pressure level was reached, the vacuum pump was switched off, and the sampler sampled air for 153 seconds. The experiment was repeated at 3 different pressure levels. Using the manifold pressure and temperature data within the vessel, logged at 3 Hz, we calculate the sampled air volume at STP using the ideal gas law as a function of sampling time. The experiments were performed at room temperature.

The pressure readings are initially constant, while the bag is expanding to its full size of 2.58 litre. Afterwards, the pressure starts to increase when air is compressed. We assume that the bag has expanded to its full size when the pressure starts to increase. Furthermore, the results allowed us to create a simple empirical model to calculate the sampled air volume as a function of sampling time at all pressure levels. It provides a useful tool to quickly estimate the expected sample size and vertical resolution during field campaigns.

Figure 3a shows the sampled air volume at STP as a function of the sampling time in seconds, for three different pressure levels (31.5 hPa, 60.8 hPa and 117.7 hPa) in the vessel. The volume is calculated with the ideal gas law, using the logged manifold pressure and temperature. The volume of the bag is estimated to be 2.58 litre. The 3 Hz pressure data was averaged

into 5-second bins, to reduce the random noise of the pressure measurements and to smooth the pressure wave induced by the stroke of the pump. The sampled air volume increases linearly with the sampling time when the bag expands to its full size during the first 20±1 second. The moment compression is required, pressure starts increasing rapidly, and this moment was found to be 20 seconds after filling initiated. Afterwards, the increase rate slows down due to reduced flow rates that result from the increasing pressure difference across the pump. The gain in sampled air volume within equal sampling time thus becomes less at longer sampling time.

Furthermore, we show the sampled air volume as a function of the vessel pressure in Figure 3b. Here the sampling times of 50, 100 and 150 seconds are arbitrarily chosen. For each sampling time, the sampled air volume is interpolated from the data shown in Figure 3a and appears to be linear to the vessel pressure. Hence, we fit the following linear equation to the derived data in Figure 3b,

$$V_{stp} = a(t)p_a \tag{2}$$

Where $p_a$ (in hPa) is the ambient pressure in the vessel, $V_{stp}$ is the sample amount in litre at STP and a is a function of the sampling time ($L_{stp}$ hPa$^{-1}$). We performed a series of linear fits for the sampling time ranging from 0 to 150 seconds at an interval of 10 seconds, and derived corresponding linear coefficients a as a function of sampling time (see Figure 3c). To model $V_{stp}$ as a function of pressure $p_a$ and sampling time $t$, the linear coefficient $a(t)$ in Equation 2 is empirically modelled using the following function:

$$a(t) = x - be^{\frac{-(t-t_0)}{\tau}} \tag{3}$$

where t is sampling time and $x$, $b$ and $\tau$ are constant parameters used for the fit. $t_0 =$19.7 seconds is the time required to fill the bag up to chamber pressure, and the model is only valid for $t >$19.7 seconds. Equation 3) is fitted using the non-linear least squares method to obtain an empirical model for the slope a(t) in Equation 2.

Combining Equation 2 and Equation 3, the sampled volume at STP can be approximated for all pressure levels ranging from 200 to 0 hPa for any chosen sampling time. The derived sample volume is shown as a function of ambient pressure or altitude in Figure 3d for the sampling time of 50, 100, 200 and 1000 seconds, respectively. The International Standard Atmosphere is used to link ambient pressure and altitude. The gain in the sample size from 200 to 1000 seconds of sampling is very small due to the saturation of the pumping capacity; however, the vertical resolution would on the other hand be compromised severely. Assuming an ascent speed of the balloon of 5 m s$^{-1}$, the corresponding vertical resolutions would be 1 km and 5 km for the sampling time of 200 s and 1000 s, respectively.

An upper limit to the amount of air samples in the MLF bag was found due to its sealing capacity. The bag's seal was observed to break when a differential pressure of ∼300 hPa between the inside and the outside of the bag is reached. The maximum allowed pressure serves as a practical limit to the sample size that can be achieved, which is presented in Figure 3b with a horizontal line, and in Figure 3d with a vertical line. During flight, the payload is usually lifted up to ∼30 km (∼10 hPa), which means the pressure inside the MLF bag can be at maximum ∼310 hPa. Conservatively, we set the maximum absolute pressure in the MLF bag during flight not higher than 280 hPa to avoid any potential loss of sample due to the burst of the bag.

The model provides a good tool to design the sampling strategy in the field. It should be noted that the simplification of the model causes uncertainties in the estimated sample size. On one hand, this model does not take the temperature in the real conditions into account. Since air in the stratosphere is usually cold e.g. 220 K, the total sampled volume at STP would be larger than the modelled, due to thermal expansion. On the other hand, the model assumes a constant upstream pressure, whereas the upstream pressure decreases during flight, and hence the total sampled volume at STP would be then smaller than the modelled.

## 5 Flights and validation

Following the laboratory experiments described above, we deployed the sampler in the field. A total of 4 flights were performed in Sodankylä, Finland (67.368°N, 26.633°E, 179 m.a.s.l.) at the Finnish Meteorological Institute's Total Carbon Column Observing Network (TCCON) facility (Kivi and Heikkinen, 2016). The facility includes a high-resolution Fourier Transform Spectrometer installation to retrieve column-averaged abundances of atmospheric constituents, gas analysers for in situ measurements and both manual and automatic radiosonde systems. The flights were performed on four different days, on 26 April, 4 – 6 September, 2017, respectively. We aimed to collect four air samples during each flight at four preset pressure altitudes. The settings of the sampling parameters are summarised in Table 4. The sampling parameters varied from flight to flight, to test the capabilities of the sampler.

The payload consisted of an AirCore, LISA Sampler, a payload positioning system that uses both Iridium and GPS/GSM positioning, a lightweight transponder and a Vaisala RS92-SGP radiosonde (Dirksen2014). This configuration allowed for a direct comparison between AirCore and sampler measurements. The AirCore used during the campaign consists of two pieces of stainless steel tubing (40 m long 1/4 in. OD and 60 m long 1/8 in. OD, wall thickness 0.01 in.), with a total weight of ∼3.6 kg. The LISA sampler package weighed 2.8 kg. After retrieval of the payload, the samples were analysed in the TCCON laboratory for $CO_2$, $CH_4$ and CO mole fractions using the same CRDS analyser as used in our laboratory, whereas the AirCore sample analysis was done on a second CRDS analyser for $CO_2$, $CH_4$ and CO mole fractions. Two different sets of calibration gases were used for the AirCore and the sampler sample analysis. Although both sets of calibration gases are ultimately on the same scales ($CO_2$: X2007, $CH_4$: X2004A and CO: X2014A), we cross checked the calibration gases on one CRDS analyser to eliminate any difference that may exist between the two sets of calibration gases.

During the flights, temperature, air pressure and pressure in the manifold were logged with a frequency of 0.8 Hz. The temperature was measured near the batteries and pump, for diagnostic purposes. Ambient atmospheric temperature was measured with the radiosonde. The logged pressure and radiosonde temperature data allowed us to quantify the sample size (subsection 5.2) and to calculate the pressure weighted mean altitude of the samples (subsection 5.1). The altitude provided by the radiosonde is used for calculation of the vertical resolution of the samples (subsection 5.2).

Fifteen samples were successfully obtained from four flights. During the first flight on 26 April 2017, only time and the start and end time of sampling were logged due to a malfunction in the datalogger. As the time stamp of the datalogger is reported in UTC, we are able to sync the sampling information from the datalogger with atmospheric measurements of temperature,

pressure and altitude from the radiosonde on the same payload. This is subsequently used to estimate the vertical resolution and the sample size using the empirical derived function in section 4. During the same flight, the AirCore datalogger failed to record any data (e.g. coil temperature, valve closing time), and the temperature data from a flight performed two days earlier, on 24 April 2017, has been used to retrieve the AirCore profiles. During the flight on 4 September 2017, the sampler was unsuccessful to take a sample at the 200 hPa pressure level, because the maximum allowed pressure in the manifold was reached during the short time between the closure of the outlet valve and opening of the sampling bag. Reversing the order of closure of the outlet valve and opening of the sampling bag fixes this problem.

## 5.1 The weighted mean sampling pressure of the samples

During sampling of each bag, the atmospheric pressure decreases as the payload ascends, and the volume flow rate drops due to a nonlinear increase of pressure in the bag. Therefore, not all atmospheric pressure levels contribute equally to the collected sample in size or mole fractions of trace gases. The integrated sample thus has an associated pressure weighted mean altitude. The contribution of each pressure level to one sample is proportional to the number of moles of air sampled at that pressure level. In general, the first 19.7 seconds of sampling contribute the most and the end of sampling contributes the least to the collected sample. When pressure and temperature within the manifold are measured, the number of moles of air at each pressure level can be computed directly, and the weight of that pressure level will be:

$$w_i = \frac{dn_i}{n} \tag{4}$$

where $dn_i$ is the number of moles of air sampled at the pressure level $p_i$, n is the total number of moles of collected air samples. $w_i$ is then the weight of the air samples collected at the pressure level $p_i$. The altitude weighted mean $\overline{p}$ can be calculated as follows:

$$\overline{p} = \Sigma w_i p_i \tag{5}$$

The first 19.7 seconds of sampling cannot be calculated directly, since $dp_i = 0$, i.e. no compression and the pressure inside the bag is the same as outside. We assign the first weight that can be calculated to the first 19.7 seconds of sampling.

The temperature of the air samples in the bag was not directly measured. For the calculation of the weighted mean sampling pressure of the samples, we assume constant temperature of the sampled air while sampling. In reality, the temperature of the air samples in the bag would be close to the ambient temperature as the bag is directly exposed to the ambient. Since the observed variability of the ambient temperature during sampling is usually less than 1 Kelvin (1-$\sigma$), the assumption of constant temperature during sampling causes insignificant uncertainty on the weighted mean sampling pressure.

## 5.2 Vertical resolution and sample size

Both the volume of the collected air samples (Figure 4a) and vertical resolution decreases (Figure 4b) with increasing altitude. The sample size achieved by the sampler is close to that estimated based on the empirical model shown in section 4. The variability of the collected sample size can be mostly explained by the different settings for the sampling time and the maximum

allowed pressure during different flights (see Table 4). Furthermore, the cold temperatures in the stratosphere result in denser air, so the observed sample size are slightly higher, especially in the lower stratosphere.

The variability in the vertical resolution is the result of three factors: 1) varying sampling time; 2) varying ascending speed; 3) varying maximum allowed pressure. The ascending speed was typically around $7 - 9 \, \mathrm{m \, s^{-1}}$ in the lower stratosphere and decreased to $4 - 5 \, \mathrm{m \, s^{-1}}$ in the middle stratosphere. The varying ascending speed accounts for the observed deviations from the otherwise linear trend in Figure 4b. In the lower stratosphere (10 to 15) $\mathrm{km}$ the maximum allowed pressure inside the bags was usually reached in a period shorter than the preset sampling time, leading to relatively high vertical resolution. In the region 10 to 15 $\mathrm{km}$ two samples deviate (4-Sep 17 $\mathrm{km}$ and 26-April 14 $\mathrm{km}$), with lower resolution, which is due to a higher ascending speed. In the middle stratosphere, the sampling time was usually the limiting factor to vertical resolution. One sample in the middle stratosphere has a relatively good vertical resolution (5-Sep, 25 $\mathrm{km}$), which is due to the relatively slow ascent speed.

While the sampler was still collecting the last sample during the flight on 6 September 2017, the balloon burst at a lower altitude (21.4 $\mathrm{km}$) than previous flights. The vertical resolution of that particular sample was estimated to be 3.1 $\mathrm{km}$ (not shown), a number much larger than that of other samples due to the fast descending speed of 16.8 $\mathrm{m \, s^{-1}}$ after burst.

### 5.3 Comparison with AirCore measurements

The vertical profiles of $CO_2$, $CH_4$ and CO mole fractions from both AirCore and LISA measurements are presented in Figure 5. For the retrieval of the AirCore profiles we refer to (Chen et al. in prep). AirCore and LISA measurements are compared based on the same pressure level. For a fair comparison, we average the AirCore profiles with the same weights that are used to calculate the weighted mean sampling pressure of the samples. The mean differences between LISA and AirCore measurements of $CO_2$, $CH_4$ and CO mole fractions are summarized in Table 5.

A relatively large difference in the $CO_2$ mole fractions (>1 ppm) between LISA and AirCore is clearly visible for the flight on 26 April 2017. The observed difference is much larger than the uncertainty caused by the drift of $CO_2$ mole fractions due to storage in the MLF bag (shown in Figure 2a) and cannot be explained by any known reasons. The differences of the $CO_2$ mole fractions for other flights are significantly smaller. The summertime stratosphere is only affected by weak diabatic stirring (Holton et al., 1995; Plumb, 2002, 2007), and can be considered relatively stable. Therefore, the flights on $4 - 6$ September 2017 can be, to a large extent, considered duplicate measurements. This is supported by the excellent agreement between the AirCore profiles of $CO_2$ and $CH_4$ mole fractions measured on those dates. The AirCore datalogger failure on 26 April 2017 may cause increased uncertainty in the altitude registration of the AirCore measurements, whereas the malfunction of the LISA datalogger during the same flight may cause increased uncertainty in the weighted mean sampling pressure of the samples. Therefore, we also calculated the mean differences excluding the flight on 26 April 2017, which decreases the mean difference in $CO_2$, but slightly increases the difference in CO.

CO mole fractions agree well during all flights, except that a small decrease with altitude was observed by LISA measurements in September 2017, but not captured by AirCore measurements. A good agreement between AirCore and LISA CO measurements is found for the flight on 26 April 2017. Besides this, an interesting CO plume at 13.5 $\mathrm{km}$ is observed by both AirCore and LISA during the flight on 5 September 2017.

## 6 Discussion

### 6.1 LISA sampler comparison with AirCore measurements

The deviation between AirCore and sampler results are on average 0.84 ppm for $CO_2$, a result that is comparable in magnitude to AirCore inter comparisons (e.g. Engel et al., 2017; Membrive et al., 2017). For methane, we find a mean deviation of 1.8 ppb, within the uncertainties of both AirCore and LISA (see subsection 3.3). CO also shows a good agreement within the measurement uncertainty of CO by the CRDS analyser. Several aspects are considered that could explain the observed differences. First of all, the altitude registration of AirCore measurements is associated with uncertainties, as outlined by (Membrive et al., 2017), especially due to the manual selection of the start and the end of AirCore sample analysis or any potential loss of air samples in case of valve malfunction, which complicates the comparison between AirCore and the sampler. Secondly, there are uncertainties associated with the calculation of AirCore weighted mean. The AirCore profile needs to be weighted for a fair comparison, since air samples at different altitudes (or pressure levels) do not contribute equally to the sampler samples. The uncertainty in altitude of the AirCore profile adds a level of uncertainty to the AirCore weighted mean. Finally, the retrieved AirCore profiles are already smoothed due to molecular diffusion and Taylor dispersion, and smearing effects in sample renewal of the cavity of the CRDS. For more information on the uncertainties associated with AirCore profiles we refer to Engel et al. (2017); Karion et al. (2010); Membrive et al. (2017).

In the first flight on 26 April, an averaged difference of $\sim 1$ ppm in $CO_2$ is observed, which cannot be explained by the associated uncertainties or by smoothing of the AirCore profile due to diffusion. The samples were taken at a distance no more than 1.5 km apart (determined from Vaisala GPS-data) in the stratosphere, with less than 1.5 hours in between LISA and AirCore sampling. Such large horizontal mole fraction gradients are not expected in the stratosphere, although stratospheric dynamics in winter show a higher degree of variability in measured trace gases. The AirCore valve did not close during the 26 April 2017 flight. This complicates the altitude registration of the AirCore, especially on the lowest part of the profile. The influence on the stratospheric part of the profile is limited, which can be seen from the large degree of agreement in $CH_4$ profiles between AirCore and LISA.

Another potential source for the bias is out-gassing from the packaging material and the balloon. As the ambient pressure decreases during ascent of the balloon flight, the desorption of trace gasses from the surface of the packaging material and the balloon occurs, which potentially influences the mole fractions of the air samples. This would, however, not explain the good agreement during the September flights. Furthermore, the inlet is located at the top of the payload and any out-gassing from the packaging material would be flushed away from the inlet during ascent.

The seasonality in tropospheric $CO_2$ that causes the difference between sampled air and its storage environment could contribute to the observed difference. The northern hemisphere winter $CO_2$ mole fractions are typically 10 ppm higher those in summer. During the storage test with low mole fractions, e.g. sample nos. 6& 7 in Figure 2, a drift of up to 0.8 ppm was observed. Therefore, a typical seasonal difference of 10 ppm could only explain a difference of 0.03 ppm in the observed $CO_2$ bias.

As seen from Table 5, the mean deviation for $CO_2$ reduces from 0.84 to 0.55 ppm, when the 26 April 2017 flight is excluded. Still, the sampler shows consistently higher $CO_2$ mole fractions than AirCore, which suggests that a small unexplained bias might exist in the LISA $CO_2$ mole fraction measurements compared to AirCore. $CH_4$ and CO on the other hand show an excellent agreement within measurement uncertainties, which suggests that no significant bias exist within the measurement uncertainties for $CH_4$ and CO.

## 6.2 Vertical resolution and sample size

The vertical resolution of the collected stratospheric air samples ranges from 500 to 1500 m, and the sample size ranges from 800 to 180 $mL_{stp}$. The vertical resolution and sample size thus outperform the AirCore. The vertical resolution and sample size of the LISA sampler is compared to the performance of AirCore sub sampled air in Table 6. Its shows that sampler outperforms the sub sample method described in Paul et al. (2016); Mrozek et al. (2016).

As mentioned previously, the vertical resolution depends on the ascending speed and the effective sampling time, and the sample size also depends on the effective sampling time. To this end, the effort in collecting more air samples by increasing the effective sampling time will compromise the vertical resolution. The vertical resolution can be improved by lowering the ascending speed and decreasing the sampling time. The pump works most effectively when the pressure difference across it is minimal. From the results shown in Figure 3d, we see that after 200 seconds of sampling the gain in sample amount decreases quickly. Therefore, the gain in sample amount, for example adding 10 extra seconds of sampling time, is small, however the decrease in vertical resolution is significant.

During the experiments described in section 4, the pump was at room temperature. The pump performance could be affected by the cold environment. First, the batteries could lose capacity and cause the power supplied to the pump to decrease. The temperature inside the thermally insulated package, where the pump is located, during flight ranged between 30 and $-15$ °C. Secondly, the diaphragm is exposed to the cold air passing through the pump. The elasticity of a rubber is temperature dependent, which could reduce the performance of the pump. On the other hand, heat is released during operation of the pump, which increases the temperature. Finally, the effect of air temperature on sample size follows the ideal gas law, and the sample size increases at low temperatures. As no experimental data is available to determine the performance of the pump at stratospheric temperature, we assume that the pump performs the same during flight as at room temperature for the calculation of the sample size.

The sample size can be increased by using an alternative pump that can deliver a higher flow rate than the current 8 $L_{stp}$ $min^{-1}$ and using sampling bags with a larger size than the current 2.58 litre. It will be mostly practical to increase the size of the sampling bag because this does not add significant weight or power consumption. An alternative more powerful pump could potentially increase the sample size, especially for the samples from high altitudes; however, it would also likely add more weight and consume more power that in turn increases the weight due to the need for more batteries.

Alternatively, to increase the amount of sample retrieved during one flight, additional bags can be considered. Currently the system is idle during several stages of the ascent as can be inferred from Figure 5. This will however be more demanding on battery power. Furthermore, care has to be taken to avoid overlapping sampling schemes i.e. sampling of a sample at altitude

P1 is still ongoing while the set-point altitude for sample two, P2 is reached. This is complicated further with variable ascent speed, that is typical for these balloon flights.

## 6.3 Uncertainty in sample amount and vertical resolution

The accuracy in sounding of the Vaisala RS92-SGP pressure sensors is 1 hPa respectively at 200 hPa, and 0.6 hPa in the range $100 - 3$ hPa, (Vaisala, 2013). The uncertainty of RS92-SGP pressure altitude is discussed in detail by Dirksen et al. (2014). The uncertainty of the vertical position of the RS92-SGP radiosonde is 20 m and hence is also the uncertainty of the pressure weighted altitude mean. Since the vertical resolution is calculated as the altitude where sampling stops minus the altitude at which sampling starts, the uncertainty in vertical resolution of the sampler is 29 m, calculated using Gaussian error
propagation.

The uncertainty in the estimated sample amount is a result of the uncertainty in pressure and temperature measurements. The pressure sensor in the manifold has an uncertainty of 2.6 hPa (Honeywell) whereas the radiosonde temperature measurements have an 1-$\sigma$ uncertainty of 0.25 K in sounding Vaisala (2013). This results in an error in estimated sample amount of 7.6 $mL_{stp}$.

Since the manifold pressure was not logged during the flight on 26 April, the pressure weighted altitude mean of the samples
had to be estimated using Equation 2 and Equation 3. As mentioned earlier, Equation 3 assumes a constant upstream pressure, which is not the case during flight where the pressure decreases. This results in errors in both the estimated sample amount and the estimated mean pressure altitude. The error in the fit parameters, is included in the calculation. The error in $p_b$ can be calculated using the uncertainties of the fit (Equation 3) and the pressure and measurements can be calculated using standard error propagation. The error in sounding of the pressure sensors is 1 hPa respectively at 200 hPa (Vaisala, 2013). The total
uncertainty after 200 seconds of sampling is 9 $mL_{stp}$, slightly higher than the effect found above.

## 6.4 Uncertainty of potential isotopic composition measurements

The stratospheric air samples can be used for analysis of isotopic composition measurements of trace gases. Here we take $CO_2$ and $CH_4$ as an example to estimate the uncertainties of isotopic composition measurements due to the storage bias (see Table 3) or the LISA-AirCore bias (see Table 5), and the estimated isotopic signatures associated with the assumed contamination
source.

The measured mole fraction ($C_m$) is the sum of the mole fraction of the original stratospheric air sample ($C_s$) and that of the contaminated air sample ($C_c$) in the bag, weighted with their respective contributions:

$$C_m = f_s C_s + f_c C_c \tag{6}$$

With $f_c + f_s = 1$ and solving for $f_c$ yields,

$$f_c = \frac{C_m - C_s}{C_c - C_s} = \frac{bias}{C_c - C_s} \tag{7}$$

given that the $C_m - C_s$ is the observed bias. The isotope composition after the mixing of the tropospheric contamination into the sample air, can be approximated with:

$$\delta_m \approx \delta_s f_s + \delta_c f_c \tag{8}$$

were $\delta_m$ is the final isotopic composition, and $\delta_s$ and $\delta_c$ represent the isotope composition of source and contamination and $f_s$ and $f_c$ are the fractional contributions to the total number of molecules after mixing. We further define the bias of the isotopic composition measurement as

$$\Delta\delta = \delta_m - \delta_s \tag{9}$$

Combining Equation 8 and Equation 9, and using $f_c + f_s = 1$ again, we derive

$$\Delta\delta = (\delta_c - \delta_s)f_c \tag{10}$$

For the calculation, we regard the mean differences between AirCore and LISA measurements (Table 5, e.g. 0.84 ppm for $CO_2$ and 1.8 ppb for $CH_4$) as the upper limit of bias induced in the stratospheric samples. Another estimate is performed based on the storage test results, that showed maximum drifts of 0.11 ppm for $CO_2$ and 2 ppb for $CH_4$, presented in Table 3.

The fraction $f_c$ can be calculated according to Equation 7 with $C_s$ being the typical stratospheric mole fraction, which is taken to be 395 ppm for $CO_2$ and 500 ppb for $CH_4$. For the contaminated air, we use typical tropospheric values of 405 ppm for the $CO_2$ mole fraction and 1800 ppb for the $CH_4$ mole fraction. The isotopic compositions $\delta_c$ and $\delta_s$ are taken from various references, and are presented in Table 7.

We can readily see that the estimated biases due to the storage are relatively small compared to the typical analytical precisions, also presented in Table 7. The estimated biases in stable isotope measurements based on the observed differences between AirCore and LISA may be significant in certain cases, but should be considered an upper estimate, since these are based on maximum differences between the troposphere-stratosphere mole fractions and isotopic composition. Hence the LISA sampler provides a viable sampling tool for useful measurements of stable isotopes in $CO_2$ and $CH_4$.

## 7  Conclusions

We have developed a new lightweight stratospheric air sampler, named LISA. The LISA sampler weighs ~2.5 kg, and is designed to collect four bag samples in the stratosphere during a balloon flight for $CO_2$, $CH_4$ and CO mole fraction measurements. Laboratory test results show that both MLF and Tedlar bags can maintain the sample mole fractions of $CO_2$, $CH_4$ reasonably well for at least 4 hours; however, we choose the MLF bag because it outperforms the Tedlar bag in the stability of both CO and water vapour. Accounting for the storage drift and analysis uncertainty, we estimate the uncertainty of the LISA sample measurements to be 0.14 ppm for $CO_2$, 2.3 ppb for $CH_4$ and 7.8 ppb for CO, respectively.

To assist the choice of the sampling strategy in terms of the sample vertical resolution and the sample size, we have evaluated the performance of the sampling pump in a pressure-controlled environment. Based on the test results, we have estimated the

expected sample size for each altitude and for each sampling time and found that the increase of the sample size is saturated around 200 seconds of sampling. A further increase of the sampling time would collect little additional air sample but decrease the vertical resolution.

The LISA sampler was successfully flown four times during balloon flights in Sodankylä, Finland, in April and September 2017, retrieving a total of 15 samples. The sample size ranges between 800 mL to 180 mL for the altitude between 12 km and 25 km, with the corresponding vertical resolution ranging from 0.5 to 1.5 km. The collected air samples were analysed for $CO_2$, $CH_4$ and CO mole fractions, and evaluated against AirCore retrieved profiles, showing mean differences of to 0.84 ppm for $CO_2$, 1.8 ppb for $CH_4$ and 6.3 ppb for CO, respectively.

The LISA sampler is thus a viable low-cost tool for retrieving stratospheric air samples, providing a complementary method to AirCore. Furthermore, The LISA sampler is advantageous to perform routine stratospheric measurements of isotopic compositions of trace gases.

*Code availability.* TEXT

*Data availability.* The data presented here is available on request. (huilin.chen@rug.nl)

*Code and data availability.* TEXT

*Sample availability.* TEXT

*Video supplement.* TEXT

*Competing interests.* TEXT

*Acknowledgements.* The authors thankfully acknowledge the help of Bert Kers, Truls Andersen, and Marc Bleeker for their help during the laboratory tests. Juha Karhu (FMI) helped with balloon launching and payload recovery which is highly appreciated. We like to thank Juha Hatakka (FMI) for maintaining the calibration gasses in Finland. This research was (in part) funded by a grant (reference number ALW-GO/15-10) from the User Support Programme Space Research. The presented AirCore flights were supported by ESA project FRM4GHG. We also acknowledge funding by the EU project RINGO, EU project GAIA-CLIM and Finnish Academy grant number 140408.

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

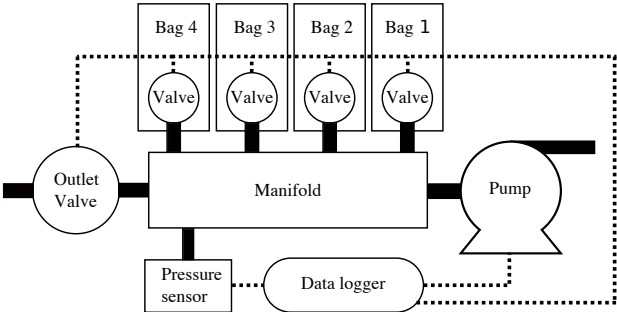

**Figure 1.** A schematic diagram of the sampler. Four bags are connected to a custom-made manifold. A small servomotor operates the screw cap combo valve. The outlet valve is the same as that of the bags, but is normally open when the sampler is idle during flight, allowing air pressure to equilibrate with outside air. Pressure inside the manifold is monitored by a pressure sensor. A datalogger is used to control all of the electronics. Electric connections are shown with dashed lines.

**Table 1.** Components used in the LISA sampler, including manufacturer and product key. The total weight is given for amounts per part. Voltage and power are presented according to manufacturer specification. The total weight for the onboard computer and sensors is given.

| Component | Company | Product key | Amount | Voltage (V) | Power (W) | Weight (g) |
|---|---|---|---|---|---|---|
| Servo motor | Hitec | HS-65HB+ | 5 | 4.8 – 6 | 1.32 | 91 |
| Pump | KNF | NMP 850.1.2 KNDC B | 1 | 24 | 10.8 | 403.6 |
| Bag (MLF) | Supelco | 30227-U | 4 | (-) | (-) | 80.4 |
| Tube Cole | Palmer | EW-95100-02 | 1 | (-) | (-) | 30 |
| Union T | Swagelok | NY-400-3 | 5 | (-) | (-) | 39 |
| Union Knee | Swagelok | NY-400-9 | 5 | (-) | (-) | 33 |
| Battery | (-) | CR123 | 10 | 3 | (-) | 166 |
| Pressure sensor | Honeywell | HSCMAND015PASA5 | 2 | | | |
| Temperature | sensor | IST 600C (100 Ω) | 1 | 7 – 12 | (-) | 87.4 |
| Datalogger | Arduino | Mega 2560 | 1 | | | |
| Battery casing | TruPower | BH-CR123A | 10 | (-) | (-) | 68.8 |

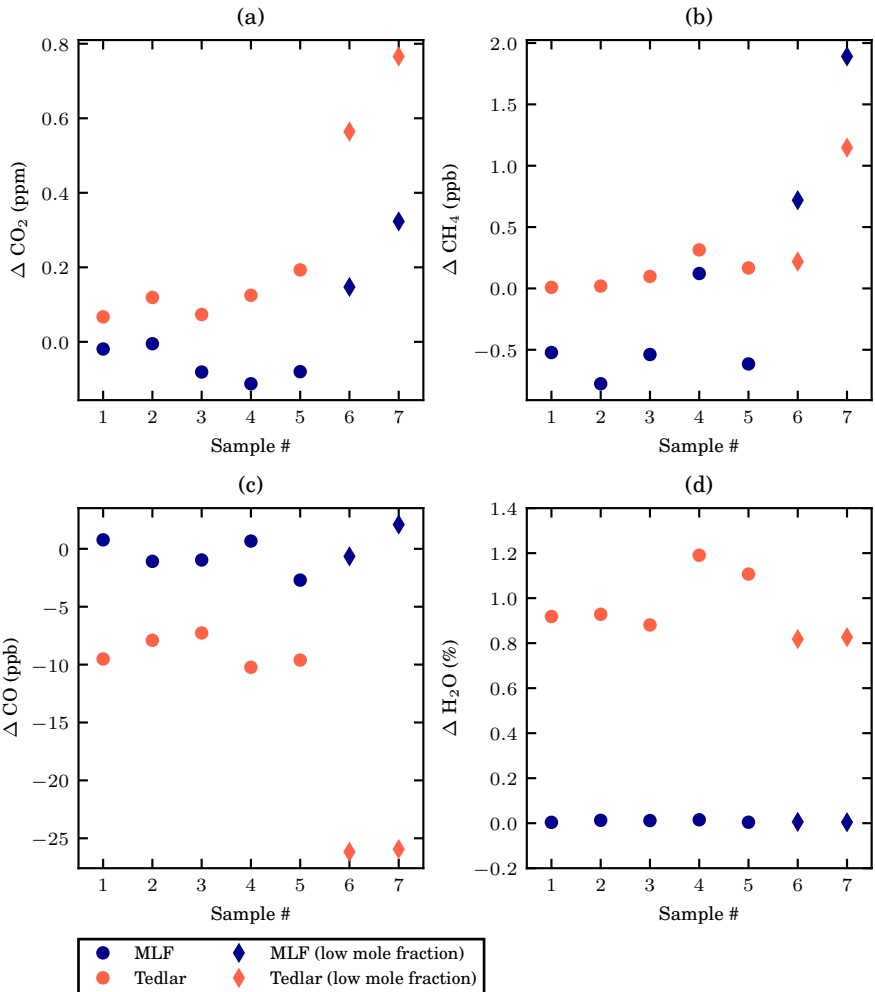

**Figure 2.** The observed drift of the mole fractions of $CO_2$ (a), $CH_4$ (b), CO (c) and H2O (d) in each of 7 samples in both Tedlar and MLF bags. The drift is defined as the difference between the measured mole fractions after 4 hours and those measured immediately after filling. For $CO_2$, the mole fractions of samples nos. $1 - 5$ are representative for stratospheric mole fractions. Samples nos. 6 & 7 contain low mole fractions and represent a typical mole fraction of stratospheric $CH_4$.

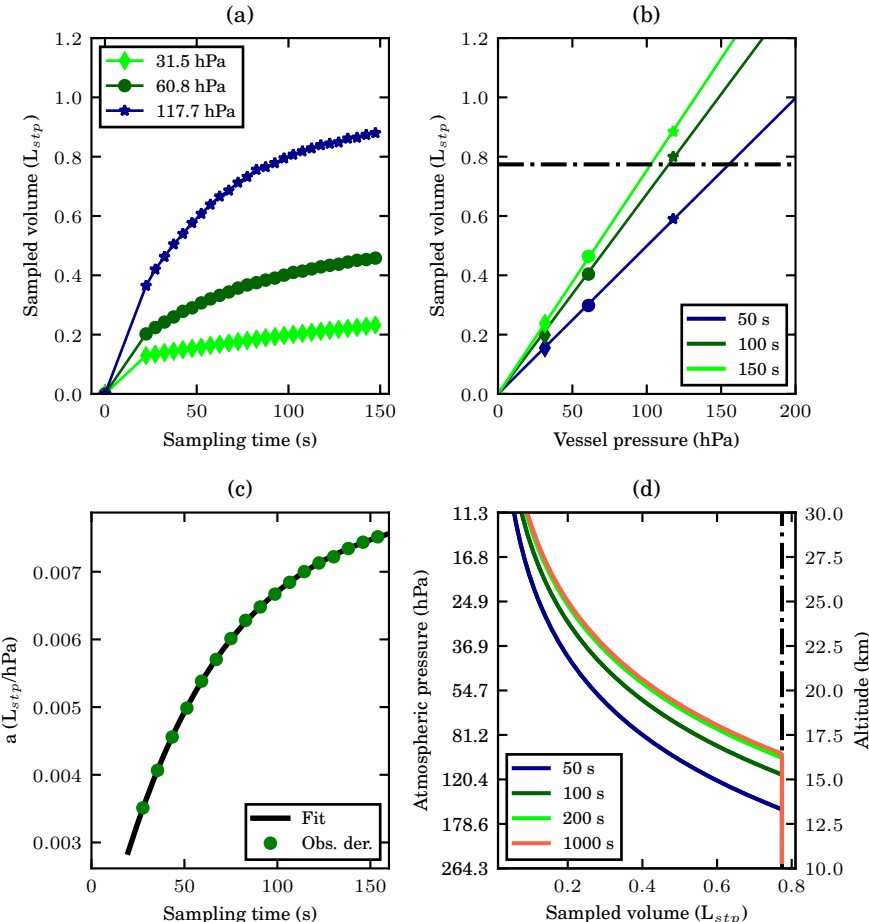

**Figure 3.** 3 a) Sample volume as a function of the sampling time in seconds. The first 19.7±0.3 seconds fill the bag up to the pressure in the vessel. The sampled volume in the first 19.7 seconds linearly interpolated starting at zero, e.g. assuming that sampled volume increases linear with time. After the first 19.7 seconds, the bag is not expanding and air needs to be compressed and the flow rate drops. b) Sampled volume in litre at STP as a function of vessel pressure (markers correspond to those presented in 3a). The sampling time of 50 seconds, 100 seconds and 150 seconds are arbitrarily chosen. The lines are a linear fit to the data as in Equation 2. For a given sampling time, the sampled amount at STP decreases linear with vessel pressure. The bags cannot withstand a pressure difference larger than 300 hPa. The practical limit is presented with a black dashed line. c) The slope a(t) (Equation 2) as a function of sampling time. The data points are derived values of (t) from the pressure data (Obs. der. is observations-derived), and the black line is the applied fit to the data, according to Equation 3, with $t_0$=19.7 seconds. The fit constants can be found in Table 2. d) Atmospheric pressure on the left and corresponding altitude on the right, as a function of modelled sampled volume. Note that for atmospheric pressure >120 and <30 hPa, as well as for sampling time >150 the depicted model relies on extrapolation of the observations. The International Standard Atmosphere is used to link pressure and altitude. The cut-of at the sample size of 0.77 $L_{stp}$ is the due to the practical fill limit of 300 hPa which consequently means that the sampling time is less. The uncertainty in volume presented in a and b is 7.6 $mL_{stp}$.

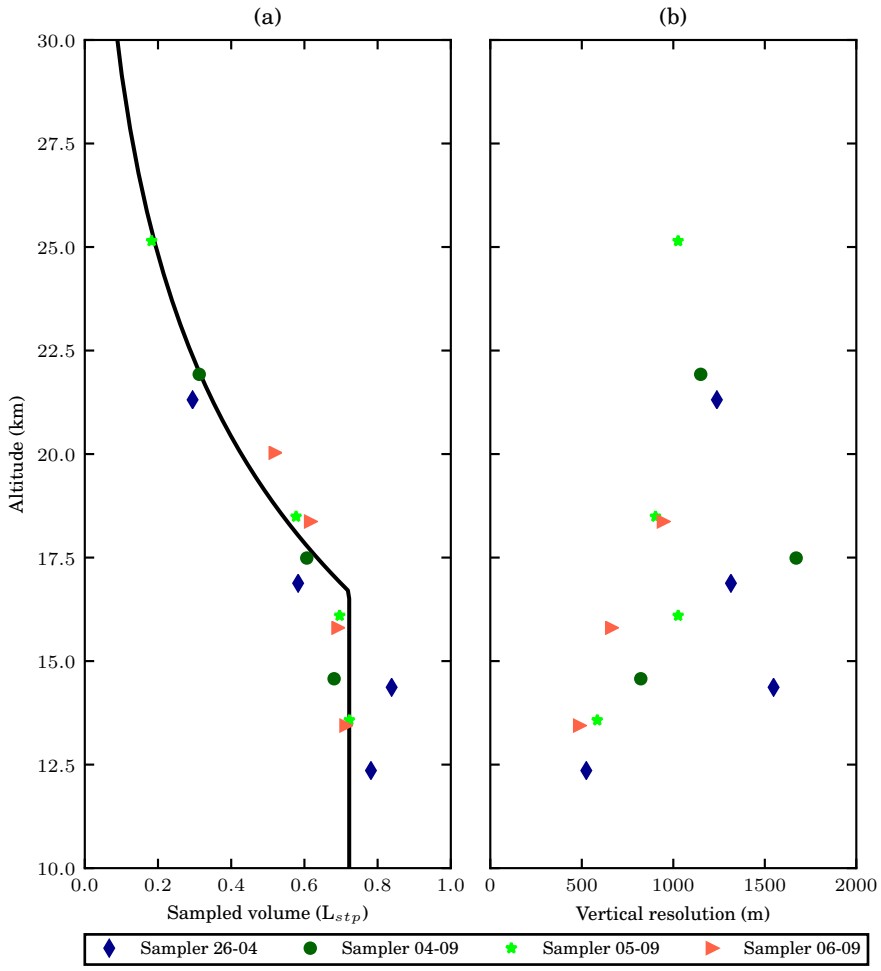

**Figure 4.** a) The altitude profile of the sample size of the collected 15 air samples. The estimated sample size with the sampling time of 200 seconds, and a maximum allowed bag pressure of 280 hPa, using the empirical relations used in section 4 is shown in blue line, the same as in Figure 3d. The uncertainty in volume presented is 7.6 mL. b) The altitude profile of the vertical resolution of the collected samples. Different colours and symbols are used to label the samples from different flights. The vertical resolution of the highest sample from the flight on 6 September 2017 is not shown as the number is abnormally large caused by fast descending speed after the burst of the balloon. The black solid line sows the expected vertical resolution, assuming an ascent speed of 5 ms$^{-1}$. Sampling time is calculated using the empirical relations discussed in section 4, with a maximum allowed pressure of 280 hPa, If this is not reached we have used the maximum sampling time of 200 seconds, which corresponds to a vertical resolution of 1 km.

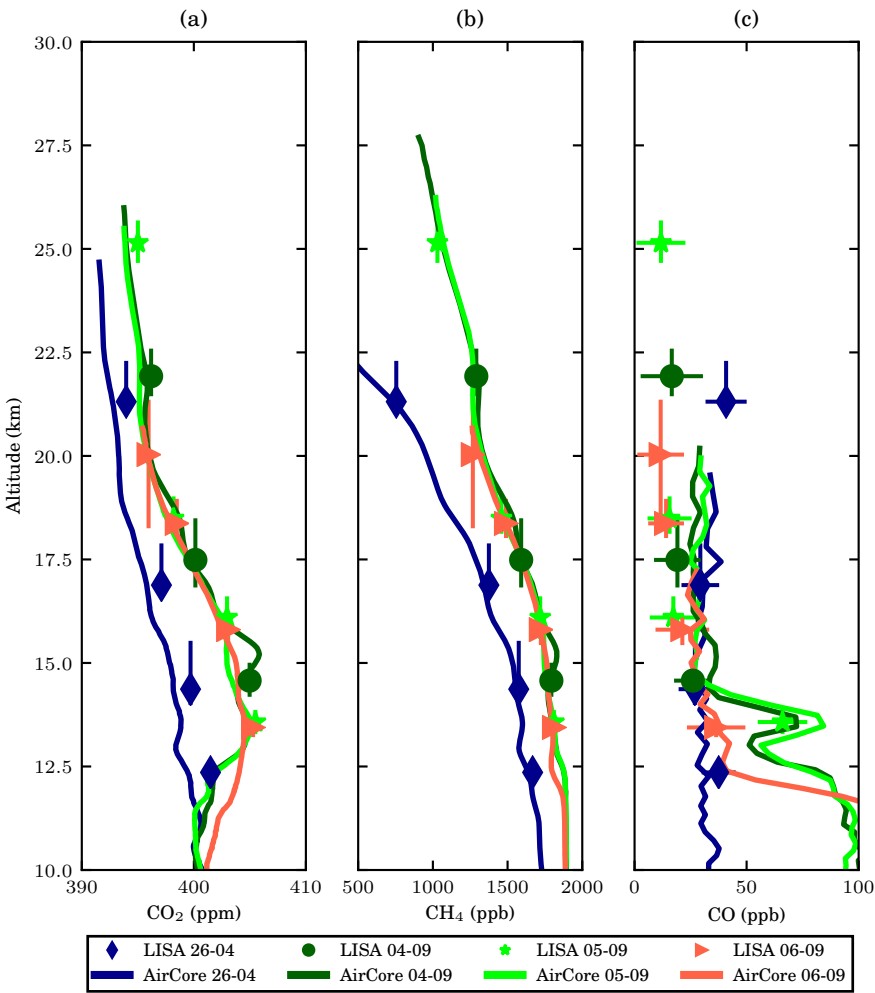

**Figure 5.** Comparison of AirCore and LISA measurements of (a) $CO_2$, (b) $CH_4$ and (c) CO mole fractions. The AirCore CO profiles are averaged in 100 m bins to smooth the relatively large noise of the measurements due to the analytical precision of 7 ppb (1-$\sigma$) of the CRDS analyser. Different colours and symbols are used to label the samples from different flights shown in the legend. All flights were performed in 2017.

**Table 2.** A total of 7 sampling bags of each type (Tedlar and MLF) were prepared with the mole fractions presented below. Sampling bag nos. 6 & 7 were filled with cylinder air and were subsequently diluted using nitrogen. The results of the CRDS analysis directly after measurement are presented for those samples.

| Sample number | | $CO_2$ (ppm) | $CH_4$ (ppb) | CO (ppb) |
|---|---|---|---|---|
| 1 | | 449.85 | 2086.2 | 260.5 |
| 2 | | 398.12 | 1969.5 | 121.5 |
| 3 | | 398.12 | 1969.5 | 121.5 |
| 4 | | 449.85 | 2086.2 | 260.5 |
| 5 | | 449.85 | 2086.2 | 260.5 |
| 6 | MLF | 127.04 | 597.5 | 74.8 |
| | Tedlar | 110.89 | 520.4 | 62.1 |
| 7 | MLF | 138.01 | 649.9 | 79.1 |
| | Tedlar | 125.89 | 591.1 | 71.7 |

**Table 3.** Uncertainty based on the different error sources for $CO_2$, $CH_4$ and CO. The total uncertainty is calculated using Gaussian error propagation. [a] The dead volume bias is estimated using a dead volume of 1.5 $mL_{stp}$, which prior to sampling is assumed to be at 50 hPa and 220 K. This volume might remain unflushed. hence the air is of tropospheric origin, with concentrations of 400 ppm $CO_2$, 1800 ppb $CH_4$ and 150 ppb CO. The total sample volume is 200 $mL_{stp}$ and has mole fractions of 395 ppm $CO_2$, 500 ppb $CH_4$ and 30 ppb CO. The bias is then calculated as the resulting deviation after mixing.

| Source | $CO_2$ (ppm) | $CH_4$ (ppb) | CO (ppb |
|---|---|---|---|
| Analyser | 0.04 | 0.2 | 7 |
| Calibration transfer | 0.07 | 1 | 2 |
| Dead volume[a] | 0.002 | 0.605 | 0.056 |
| Storage drift | 0.11 | 2.0 | 2.7 |
| Total | 0.14 | 2.3 | 7.8 |

**Table 4.** Preset sampling parameters. Sampling is completed after either the maximum pressure in the manifold or the maximum sampling time is reached. P1-P4 are the preset targeted pressure altitudes.

| Date | Maximum sampling time (s) | Maximum pressure (hPa) | P1 (hPa) | P2 (hPa) | P3 (hPa) | P4 (hPa) |
|------|---------------------------|------------------------|----------|----------|----------|----------|
| 26-Apr-2017 | 250 | 250 | 200 | 150 | 100 | 50 |
| 04-Sep-2017 | 180 | 275 | 200 | 150 | 100 | 50 |
| 05-Sep-2017 | 220 | 280 | 170 | 120 | 80 | 30 |
| 06-Sep-2017 | 250 | 280 | 170 | 120 | 80 | 50 |

**Table 5.** Comparison of $CO_2$, $CH_4$ and CO mole fractions between AirCore and LISA measurements. The difference is calculated as LISA$-$AirCore. The correlation coefficient between LISA and averaged AirCore is also presented. [a] Excluding the April 26 flight.

| Species | Mean$\pm\sigma$ | $R^2$ | Mean$\pm\sigma^a$ | $R^{2a}$ |
|---------|-----------------|-------|-------------------|----------|
| $CO_2$ (ppm) | 0.84$\pm$0.47 | 0.93 | 0.55$\pm$0.13 | 0.97 |
| $CH_4$ (ppb) | 1.8$\pm$16.2 | 0.99 | $-5.1\pm$13.1 | 0.99 |
| CO (ppb) | 6.3$\pm$6.6 | 0.58 | $-9.2\pm$5.2 | 0.59 |

**Table 6.** Comparison of the vertical resolution and sample size between the LISA sampler and samples sampled from AirCore. References: A (Mrozek et al., 2016); B (Paul et al., 2016); C this study.

| Altitude (km) | Method | Resolution (m) | Sample size (mL) | Reference |
|---------------|--------|----------------|------------------|-----------|
|    | AirCore | 800 | 25 | A |
| 12 | AirCore | 1000 | 50 | B |
|    | LISA | 580 | 720 | C |
|    | AirCore | 1500 | 25 | A |
| 15 | AirCore | 2000 | 50 | B |
|    | LISA | 820 | 680 | C |
|    | AirCore | 2000 | 25 | A |
| 20 | AirCore | 3000 | 50 | B |
|    | LISA | 1100 | 312 | C |
|    | AirCore | 3000 | 25 | A |
| 25 | AirCore | 5000 | 50 | B |
|    | LISA | 1000 | 182 | C |

**Table 7.** Expected bias in stable isotope measurements on samples obtained by LISA, due to the limited accuracy of the LISA sampler. Typical values for the troposphere and stratosphere are taken from the indicated references: A) (Trolier et al., 1996) B) (Mrozek et al., 2016) C) (Nisbet et al., 2016) D) (Bergamaschi et al., 2001) E) (Aoki et al., 2003) and F) (Röckmann et al., 2011). Reported measurement reproducibility's, Re, for stratospheric air are also provided. $\delta^{13}C$ and $\delta^{18}O$ values are with respect to Vienna Pee Dee Belemnite (VPDB) and $\Delta^{17}O$ and $\delta^{2}H$ are with respect to Vienna Standard Mean Ocean Water (VSMOW). $f_c$ was calculated using a source value 395 ppm ($CO_2$) and 500 ppb ($CH_4$), and contamination mole fraction of 405 ppm ($CO_2$) and 1800 ppb ($CH_4$). For $f_{c1}$ contamination values of 0.84 ppm ($CO_2$) and 1.8 ppb ($CH_4$) based on LISA AirCore observed mean bias, resulting in $\Delta\delta_1$; For $f_{c2}$ the maximum observed drift (Figure 2) of 0.11 ppm ($CO_2$) and 2 ($CH_4$) are used, resulting in $\Delta\delta_2$.

| | | | | LISA-AirCore | | Storage drift | |
|---|---|---|---|---|---|---|---|
| Species | $\delta_c$ ‰ | $\delta_s$ ‰ | Re ‰ | $f_{c1}$ | $\|\Delta\delta_1\|$ ‰ | $f_{c2}$ | $\|\Delta\delta_2\|$ ‰ |
| $\delta^{13}C(CO_2)$ | -7.5 (A) | -8.4 (E) | 0.02 (E) | | 0.08 | | 0.01 |
| $\delta^{18}O(CO_2)$ | -2 (A) | 12 (E) | 0.05 (E) | 0.084 | 1.18 | 0.011 | 0.15 |
| $\Delta^{17}O$ ($CO_2$) | 0 (B) | 7 (B) | 0.2 (B) | | 0.59 | | 0.08 |
| $\delta^{13}C$ ($CH_4$) | -47 (C) | -20 (F) | 0.7 (F) | 0.0014 | 0.04 | 0.0015 | 0.04 |
| $\delta^{2}H(CH_4)$ | -85 (D) | 190 (F) | 2.3 (F) | | 0.38 | | 0.42 |