# Peer review of "LISA: a lightweight stratospheric air sampler"

_Atmospheric Measurement Techniques, 2018_

## Referee Comment (RC1) · Anonymous Referee #1 · 28 Feb 2018

The paper by Hooghiem presents a new technique for sampling of stratospheric air which may be well suited to add to the available techniques of cryogenic whole air sampling and AirCore. The paper is well written, although the introduction is a bit like a collection of information on sampling techniques and the use of stratospheric trace gas measurements, but lacks a clear argumentation line. The subject is very much in line with the scope of AMT.

I have a range of minor suggestions/questions and one major observation. The major observation explained below should be clarified before publication.

Major issues My main issue is the discussion of the differences between AirCore and LISA. First, it would be extremely valuable to see the paper by Chen et al., describing the data evaluation of the AirCore system used here. More importantly, I have doubts

[Figure]

about the way that the fractional contribution of a contamination source (fc) is derived in section 6.4. In my view, calculation of fc from observed CO2 [CO2] and deviations between this observation and the expected stratospheric value ([CO2]s) should be calculated as follows (values in parenthesis are mixing ratios):

[CO2] = fs * [CO2]s + fc * [CO2]c = (1-fc) * [CO2]s + fc * [CO2]c

With the subscripts as defined in the paper. Solving this for fc yields:

fc = ([CO2] – [CO2]s) / ([CO2]c-[CO2]s)

If I assume that ([CO2] – [CO2]s) is the difference of 0.84 ppm CO2 between observed CO2 in LISA and in AirCore, I need to make an assumption on ([CO2]c-[CO2]s) to calculate fc. If we assume that -[CO2]s is stratospheric CO2 at about 390 ppm (actually 395 is more realistic), and that the contamination is from tropospheric CO2 with a mixing ratio of around 405 ppm, then I calculate fc to be 0.84/(405-395) = 0.084, and not 0.0021 as derived in Table 6.

This might actually also explain the differences observed in April flights with respect to the September flights. During Fall NH CO2 is expected to be much smaller, thus the difference between contaminant and actual stratospheric mixing ratio is much lower than during spring (when tropospheric CO2 may be up to 10 ppm higher). This is also in line with the much higher deviations found in the laboratory experiments when having larger concentration differences.

Minor/specific observations.

p.3. l. 1: the Engel et al. trend is only representative for the mid-latitudes of the Northern Hemisphere, above 24 km altitude.

p.3. l. 3.: A reference from 1983 may not be very good to point to current deficits in GCMs.

p.3. l. 26.: could you be more specific on the allowed weight?

[Figure]

p. 3. L. 29.: as LISA samples during ascent, has possible outgassing of $CO_2$ from the Styrofoam be considered?

p. 5 . section 3.1. Could you comment on how dry the test samples were? This will make a very large difference, especially for $CO_2$.

p.6. l. 13.: How were the bags preconditioned. Why was $N_2$ used and not ambient air?

p. 7. Section 4.: At which temperature was the flow characterized? Strat. Temperatures are much lower, which may influence pump performance quite strongly.

p. 7. L 8 sample, not samples

p. 7. L. 11 (and elsewhere in the manuscript): please be consistent in the use of L vs. L STP.

p.8. l. 25: please restrict this to 30 hPa, as the flow was not measured at lower pressures.

p. 9. L. 6.: Is burst pressure of the bag temperature dependent?

p. 9. L. 10: 240 K is actually very warm for the stratosphere.

p. 12. L. 17 l. 27: please be consistent in using only one value for the deviations. If the value of 0.84 ppm is used, this is much larger than stated in Engel et al., 2017. Have the authors made an uncertainty estimate for the flight on April 26, considering that no AirCore coils temperature is available?

p. 12 l. 26: It is not only molecular diffusion.

p. 12/13. Section 6.1.: This whole section lacks a conclusion. First, which is the best estimate of the deviations, and second what could cause the strong deviations during the April flight (see also my major comment above). Have pollution/outgassing been considered to explain the differences?

p. 13. L. 14.: this sentence is odd. How should a prolonged sampling time result in

increased vertical resolution?

p. 14.l. 7.: this sentence is wrong, uncertainty is mentioned twice.

p. 14. Section 6.4. see major comment above. I believe that the contaminating fraction may be much higher.

p. 15. L. 24: this is not up to 0.84 ppm. 0.84 ppm is actually an average deviation if all samples are considered.

Figure 2: the colours for the different bags types are virtually undistinguishable.

Figure 3: I wonder if all panels are needed here. I suggest removing panel b. Table 6: please explain the different columns on the table heading, not only in the text.

[Figure]

---

## Referee Comment (RC2) · Anonymous Referee #2 · 15 Mar 2018

General comments

The authors present a new stratospheric sampling system that can fill a niche between current techniques, as it provides larger air samples with better vertical resolution than AirCore while operating with much lower cost and payload weight than typical cryogenic whole air systems. The manuscript is suitable for AMT and well-written and should be published after addressing the following comments and requests. In particular, a possible error in the calculation of uncertainty for isotopic analysis should be resolved.

Major issue: Section 6.4

I agree with Reviewer 1 that the underlying assumptions used to estimate fraction of sample from contamination (fc) are incorrect. The ultimate source of the error seems

to derive from the expression used to define fc =bias/([X] + bias). Here I presume that [X], defined by the authors as "typical mole fraction," is therefore meant to be $\approx$[X]m, the measured mixing ratio.

If we rearrange equation (7) to $\delta$m = $\delta$s − $\Delta\delta$ and rewrite for mixing ratio as [X]m = [X]s − $\Delta$X, where [X]s is the stratospheric mixing ratio and $\Delta$X is the bias, then substitute this into the equation above, we have fc = $\Delta$X / ($\Delta$X − [X]s + $\Delta$X) = $\Delta$X / [X]s. But since the authors also define fc = $\Delta$X / ([X]s – [X]c) in equation (8), this implies that the authors are assuming the mixing ratio of the contamination is 0 when they calculate the fc on pg 14, line 28. Instead, they should use measured or typical mixing ratios, nominal 400 ppm and 1800 ppb for CO2 and CH4, respectively. This would result in roughly the same estimate as Reviewer 1 has calculated.

An alternative method to estimate fc would be to solve equation (8) for both CO2 and CH4 simultaneously. Since the authors already have estimates for $\Delta\delta$ and $\delta$s stated (pg 14, lines 26-28), this results in two equations with two unknowns, which can then be solved trivially. This would be a worthwhile exercise to confirm the estimate of fc provided by Reviewer 1.

Finally, it is not clear to me why the authors don't use the storage test results, in addition to the ambient data, to estimate fc. This should either be performed, or the authors should explain why this analysis is not valid.

Additional general comments

I agree with Reviewer 1 that the introduction is a bit unfocused and broad. I think a more focused discussion of the literature with regards to the particular challenges of making sufficiently accurate and precise mixing ratio measurements in the stratosphere would be more useful. See the introduction offered by Membrive et al., 2017 for an example.

The manuscript presents multiple tables of information (e.g. Table 2 is derived fit co-efficients, Table 3 is instrument operational settings) that would be better suited in

supplemental materials, as this information is not critical to main discussion here). Meanwhile, there is no table provided that summarizes the instrument specifications (e.g. weight, power requirements, sample resolution at stated altitude) and comparison to the AirCore and other systems. I would be appreciative of such a summary table.

Additional comments

p. 1, line 11 - The abstract discusses the stability tests, which include $H_2O$ mole fractions. However, the rest of the abstract only discuss $CO_2$, $CH_4$ and CO mole fractions. A phrase noting the significance of $H_2O$ (interferent?) would be appreciated.

p. 1, line 29 – "Stratospheric changes in ozone and water vapour levels" is awkwardly phrased. How about "Changes in stratospheric ozone and water vapour levels"?

p. 2, line 1 – "stratospheric air up to 35 km" Please define this value (e.g. ASL).

p. 2, line 13 – "remarkable scientific efforts" I would avoid using words like "remarkable" without a substantial defense of this term.

p. 2, line 15 – Introducing mean age of air is unnecessary here unless it is discussed later in the text

p. 3, line 8 – "AirCore does not provide large sample amount" Please quantify this statement.

p. 4, line 1 – "a diaphragm pump (KNF, product no. NMP 850.1.2 KNDC B)" Please provide a description of wetted surfaces for this pump.

p. 5, line 20 – Storage test results. Can the authors take advantage of the results of these tests to separate leak and permeability effects for all species? The discussion of the water vapor experiment is nicely done and touches on observed changes in CO; can this be extended?

p. 6, line 11 – "we observed an offset" Please describe which direction these offsets

were (increase or decrease). As a reader, I wondered if this effect could have been a cause of the CO2 offset observed during the April flight discussed later

p. 6, line 24 – "We do not include any sampling error in the presented evaluation, although it might be significant for high-precision measurements, especially for CO in the stratosphere when the ozone concentration is high." This statement was ambiguous to me. Please explain exactly what is meant by "sampling error" and the significance of ozone concentration.

p. 6, line 28 – "However, detailed information about storage conditions are required to correct for the drift, which is usually unavailable in the field." Why wouldn't this be available? The detector used for analysis was in the field (see p. 9, line 27) so this data may have been collected for the samples presented here. Please explain if the data is available and whether it should be collected in future work.

p. 7, line 12 – "The performance of the small diaphragm pump is to our best knowledge not previously investigated under the atmospheric conditions in the stratosphere, e.g. at low-temperature and low-pressure conditions." This is a nice presentation of this experiment. Was the pump subject to stratospheric temperatures? The manufacturers specification is for ambient temperature between +5°C and +40°C. Any insight that can be provided about the pump performance at stratospheric temperatures would be appreciated.

p. 8, line 6 – "19.7 seconds." How precise is this value? I'm a bit surprised that the pumping speed would be so consistent at varying pressure. There are no data points shown in Figure 3 prior to 20 sec, which makes this statement a bit hard to evaluate. The uncertainty of this value under operating conditions should be discussed with respect to the modeled behavior that follows.

p. 8, line 25 – "The sampled volume at STP can be modelled for all pressure levels ranging from 200 to 0 hPa" It seems inappropriate to model to 0 hPa, given that the data collected ends at 30 hPa.

p. 9, line 5 – "To be on the safe side," should be rephrased "Conservatively,"

p. 11, line 10 – "the vertical resolution increases" The vertical resolution decreases since the vertical height increases. There are several significant deviations from the relationship between altitude and vertical resolution, as shown in Figure 4b: 26-Apr at 150 hPa, 4-Sep at 100 hPa and 5-Sep at 30 hPa. Could these data points be explained in the context of the discussion of the variability in vertical resolution starting at p. 11, line 14?

p. 11, line 25 – While the vertical profiles of the mixing ratios are shown in Figure 5, I don't see the actual values presented. Could this be added to Table 3 or as a Supplemental Table (along with individual uncertainties of both mixing ratio and vertical height)?

p. 12, line 8 - While there are significant differences between the LISA and AirCore retrieved mixing ratios for $CO_2$, the $CH_4$ and CO mixing ratios show good agreement. Can this allow the authors to make a statement about the validity of their assumptions for estimating altitudes and pressures for the failed data loggers? And does this indicate that the offset $CO_2$ measurement is a result of the measurement of mixing ratio rather than altitude? This comment is also for p. 12 line 30 to p. 13, line 2.

p. 12, line 17 – "0.5 ppm or 0.13%" I'm not sure that the percentage reported here is appropriate, as it implies greater analytical precision that is achieved. If the $CO_2$ mixing ratio varies only 20 ppm or less between the troposphere and the top of the sampling region, then 0.5 ppm is a more significant deviation. I would suggest simply removing these percentages.

p. 13, line 13 – "decreases fast." Should be "decreases quickly." In the following sentence, "the gain in sample amount, for example adding 10 extra seconds of sampling time, does not increase" is not correct, but rather the gain in sample amount is very small in comparison to the decrease in vertical resolution. This should be restated.
p. 13, line 16 - As an aside, I would suggest additional bags as an alternative to larger bags, as the altitude profiles presented in Figure 5 indicate that there are periods during the ascent when no sample is collected, i.e. the system is sitting idle.

p. 13, line 28 – It is not clear why the uncertainty in the sample amount is important. Could the authors explain the importance of this?

p. 23 – Figure 3b. I found the reuse of marker shapes between 3a and 3b confusing (I was looking for a relationship between the same marker shape in both figures). Could the authors change the marker shapes in Figure 3b to avoid this? For Fig 3c, it is not clear if the units on the vertical axis use volumetric liters or standard liters. This should be standard liters for consistency. Also, in Fig 3d please extend the vertical line at 0.76 L for the full vertical height of the figure, as was done in Fig 3b.

p. 25 – Figure 4 and Figure 5. As a color-blind reader I cannot distinguish between the colors used for the 26-Apr and 4-Sep flight data. Could the 26-Apr color be changed to a different color (e.g. blue)?

p. 27 – Table 6. Please define the acronyms VPDB and VSMOW.

Reference: Membrive, O., Crevoisier, C., Sweeney, C., Danis, F., Hertzog, A., Engel, A., Bönisch, H., and Picon, L.: AirCore-HR: a high-resolution column sampling to enhance the vertical description of $CH_4$ and $CO_2$, Atmos. Meas. Tech., 10, 2163-2181, https://doi.org/10.5194/amt-10-2163-2017, 2017.

---

## Author Comment (AC1) · 13 Jul 2018

Our reply to the reviewers' comments and a marked-up version of the revised manuscript can be found in the attached zip file.

Please also note the supplement to this comment:
https://www.atmos-meas-tech-discuss.net/amt-2018-23/amt-2018-23-AC1-supplement.zip

---

## Author Response (AR1)

**Reply to Interactive comment of Referee #1 on "LISA: a lightweight stratospheric air sampler" by Joram J. D. Hooghiem et al.**

We thank the two reviewers for their helpful and detailed comments on our manuscript. We have addressed the major comments from both reviewers regarding our derivation of the bias estimation in isotopic composition measurements and revised the introduction according to the comments of both reviewers. Below, a detailed point-to-point reply and a revised version of our manuscript with track changes are given.

*The paper by Hooghiem presents a new technique for sampling of stratospheric air which may be well suited to add to the available techniques of cryogenic whole air sampling and AirCore. The paper is well written, although the introduction is a bit like a collection of information on sampling techniques and the use of stratospheric trace gas measurements but lacks a clear argumentation line. The subject is very much in line with the scope of AMT.*
*I have a range of minor suggestions/questions and one major observation. The major observation explained below should be clarified before publication.*

*Major comments:*

*Major issues My main issue is the discussion of the differences between AirCore and LISA. First, it would be extremely valuable to see the paper by Chen et al., describing the data evaluation of the AirCore system used here. More importantly, I have doubts about the way that the fractional contribution of a contamination source (fc) is derived in section 6.4. In my view, calculation of fc from observed CO2 [CO2] and deviations between this observation and the expected stratospheric value ([CO2]s) should be calculated as follows (values in parenthesis are mixing ratios):*
*[CO2] = fs \* [CO2]s + fc \* [CO2]c = (1-fc) \* [CO2]s + fc \* [CO2]c With the subscripts as defined in the paper. Solving this for fc yields: fc = ([CO2] – [CO2]s) / ([CO2]c-[CO2]s)*
*If I assume that ([CO2] – [CO2]s) is the difference of 0.84 ppm CO2 between observed CO2 in LISA and in AirCore, I need to make an assumption on ([CO2]c-[CO2]s) to calculate fc. If we assume that -[CO2]s is stratospheric CO2 at about 390 ppm (actually 395 is more realistic), and that the contamination is from tropospheric CO2 with a mixing ratio of around 405 ppm, then I calculate fc to be 0.84/(405-395) = 0.084, and not 0.0021 as derived in Table 6.*

We thank the reviewer for pointing out the calculation error. We mistakenly assumed that the contamination results from one-way diffusion of ambient $CO_2$ into the bag. If the observed bias would be entirely caused by diffused contamination air, the fraction of the contamination air is indeed about 8 percent. However, the observed bias is unlikely caused by diffusion alone based on our laboratory experiment results, where we found much smaller biases (~0.1 ppm for $CO_2$ and ~2 ppb for $CH_4$), as shown in Figure 2.

We have further thoughts on the diffusion process. According to the Fick's law, the diffusion depends on the diffusivity and the concentration gradient, which may be different for different species, e.g. the concentration gradient for $O_2$ and $N_2$ across the bag's material is negligible, and the concentration gradients for $CO_2$ and $CH_4$ depend on their ambient concentrations. We have updated the manuscript with the following analysis of the problem.

[revised manuscript text omitted]

*This might actually also explain the differences observed in April flights with respect to the September flights. During Fall NH CO2 is expected to be much smaller, thus the difference between contaminant and actual stratospheric mixing ratio is much lower than during spring (when tropospheric CO2 may be up to 10 ppm higher). This is also in line with the much higher deviations found in the laboratory experiments when having larger concentration differences.*

A 0.8 ppm difference was observed during the storage test (Fig. 2) when a rough 250-ppm (assuming room air of about 405 ppm) concentration difference existed across the bags material over 4 hours. According to this storage test result, a seasonal cycle of 10 ppm in CO2 between winter and summer would cause a difference of 10/250x0.8=0.032. Hence the observed increase of 1 ppm observed form the AirCore LISA difference cannot be fully attributed to the seasonal variability. This is now explicitly mentioned in the revised manuscript.

*"The seasonality in tropospheric $CO_2$ that causes the difference between sampled air and its storage environment could contribute to the observed difference. The northern hemisphere winter $CO_2$ mole fractions are typically 10 ppm higher those in summer. During the storage test with low mole fractions, e.g. sample nos. 6&7 in Figure 2, a drift of up to 0.8 ppm was observed. Therefore, a typical seasonal difference of 10 ppm could only explain a difference of 0.03 ppm in the observed CO2 bias."*

We would like to note that citing a publication in preparation, e.g. Chen et al. in prep, is in line with the guidelines of AMT. Nevertheless, we have added a sentence to provide more details about the AirCore measurements.

*"The AirCore used during the campaign consists of two pieces of stainless steel tubing (40 m long ¼ in. OD and 60 m long 1/8 in. OD, wall thickness 0.01 in.), with a total weight of ~3.6 kg."*

Minor/specific observations:

*p.3. l. 1: the Engel et al. trend is only representative for the mid-latitudes of the Northern Hemisphere, above 24 km altitude.*

We have added this in the revised version.

*"...however no significant change in the strength of the BDC in the northern hemisphere at mid latitudes was detected (Engel et al., 2009, 2017)."*

*p.3. l. 3.: A reference from 1983 may not be very good to point to current deficits in GCMs.*

We have added a more recent reference (Gerber et al., 2012).

*"In spite of all the efforts to make observations of stratospheric tracers, GCM's remain poorly constrained (Gerber et al., 2012), a problem already pointed out several decades ago (Ehhalt et al., 1983)."*

*p.3. l. 26.: could you be more specific on the allowed weight?*

The total payload of a weather balloon typically ranges between 0.2-12 kg, which represents the range of the weight from radiosondes to medium-weight scientific instrumentation. We have changed the original sentence in the revised version as

*"... typically ranges between 0.2-12 kg..."*

*p. 3. L. 29.: as LISA samples during ascent, has possible outgassing of $CO_2$ from the Styrofoam be considered?*

Yes, this has been considered. Currently, we don't have data on outgassing under stratospheric conditions.

*"Another potential source for the bias in the $CO_2$ mole fractions is outgassing from the packaging material and balloon. As the balloon ascents the surrounding pressure decreases and gasses desorb from the surface of the packaging material and balloon, which potentially influences the mole fractions. This would, however, not explain the good agreement during the September flights opposed to the April flights. Furthermore, the inlet is located at the top of the payload and any outgassing is flushed away from the inlet during ascent."*

*p. 5. section 3.1. Could you comment on how dry the test samples were? This will make a very large difference, especially for CO2.*

Test samples were prepared with air from cylinders that typically contain less than 0.03% H2O. This value has been added to the revised version. The direct measurements yielded values

below 0.06 % for the MLF bags and 0.12 % for the Tedlar bags, but since we used water vapour correction functions we did not add the absolute water content to the manuscript.

*"...with dry air (<0.03% $H_2O$) from a cylinder"*

p.6. l. 13.: How were the bags preconditioned. Why was N2 used and not ambient air?

A small discussion on the preconditioning procedure has been added in the revised version. In principle, ambient air could be used as well.

*"The bags were filled with $N_2$ from a cylinder and subsequently evacuated with a vacuum pump, prior to filling with test sample. In principle ambient air can be used to flush the bags, as long as it is dry."*

p. 7. Section 4.: At which temperature was the flow characterized? Strat. Temperatures are much lower, which may influence pump performance quite strongly.

Indeed, no laboratory tests were performed at stratospheric temperatures. The pump performance could be affected by the cold environment. First, the batteries could lose capacity and cause the power supplied to the pump decreasing. The temperature inside the thermally insulated package, where the pump is located, during flight ranged between 30 and -15 ˚C. Secondly, the diaphragm is exposed to the cold air passing through the pump. The elasticity of a rubber is temperature dependent, which could reduce the performance of the pump. On the other hand, heat is released during operation of the pump, which increases the temperature. Finally, the effect of air temperature on sample size follows the ideal gas law, and the sample size increases at low temperatures. As no experimental data is available to determine the performance of the pump at stratospheric temperature, we assume that the pump performs the same during flight as at room temperature for the calculation of the sample size.

We have clarified in the manuscript with the following text in section 6

*"During the experiments described in Section 4, the pump was at room temperature. The pump performance could be affected by the cold environment. First, the batteries could lose capacity and cause the power supplied to the pump to decrease. The temperature inside the thermally insulated package, where the pump is located, during flight ranged between 30 and -15 ˚C. Secondly, the diaphragm is exposed to the cold air passing through the pump. The elasticity of a rubber is temperature dependent, which could reduce the performance of the pump. On the other hand, heat is released during operation of the pump, which increases the temperature. Finally, the effect of air temperature on sample size follows the ideal gas law, and the sample size increases at low temperatures. As no experimental data is available to determine the performance of the pump at stratospheric temperature, we assume that the pump performs the same during flight as at room temperature for the calculation of the sample size."*

p. 7. L 8 sample, not samples

The typo has been corrected in the revised version.

*"...of air sample collected into…"*

p. 7. L. 11 (and elsewhere in the manuscript): please be consistent in the use of L vs. L STP.

We have made sure that the manuscript is now consistent with the use of L at STP, except for the volume of an object, e.g. the volume of the bag is 2.58 litre.

*p.8. l. 25: please restrict this to 30 hPa, as the flow was not measured at lower pressures.*

We now clarify in the figure caption that the presented curves are modelled based on the fitted parameters. The extrapolated curves are informative, although they may contain larger uncertainties than those within the tested range. Therefore, we would like to keep the curves outside the tested range, but have made the point clear in the revised version.

*"...Atmospheric pressure on the left and corresponding altitude on the right, as a function of modelled sample volume. Note that for atmospheric pressure >120 and <30 hPa, as well as for sampling time >150 s the modelled results rely on extrapolation of the observations."*

*p. 9. L. 6.: Is burst pressure of the bag temperature dependent?*

No experiments have been performed to show whether the burst pressure of the bag is temperature dependent. All bags survived the four flights. Two bags burst at room temperature at the pressure gradients across the bags of 312 and 331 hPa, respectively. The burst temperature might be temperature dependent; however, we think that the variation of the quality of the bags explains the different burst pressures at room temperature.

*p. 9. L. 10: 240 K is actually very warm for the stratosphere.*

Changed to 220 K, which is in line with the standard atmosphere.

*"...cold e.g. 220 K, the total sampled volume at"*

*p. 12. L. 17 l. 27: please be consistent in using only one value for the deviations. If the value of 0.84 ppm is used, this is much larger than stated in Engel et al., 2017. Have the authors made an uncertainty estimate for the flight on April 26, considering that no AirCore coils temperature is available?*

In the manuscript we have made sure that it is clear which deviation are discussed. The effect of coil temperature on the profile is relatively small.

*p. 12 l. 26: It is not only molecular diffusion.*

Indeed, we have revised the sentence to include also the effects from Taylor dispersion and air mixing in the cavity of the CRDS analyser. We also state that the altitude registration in pressure coordinates is sensitive to user input.

*"Two aspects contribute to errors in the AirCore weighted mean. First of all, the AirCore profile needs to be weighted, since not all the pressure levels contribute equally to the sampler samples. The uncertainty in altitude of the AirCore profile adds a level of uncertainty to the AirCore weighted mean. Secondly, the retrieved AirCore profiles are already smoothed due to molecular diffusion and Taylor dispersion, and smearing effects in sample renewal of the cavity of the CRDS. Moreover, the AirCore profile suffers from uncertainty in altitude registration.*

*For more information on the uncertainties associated with AirCore profiles we refer to Engel et al.(2017), Karion et al. (2010) and Membrive et al. (2017).”*

*p. 12/13. Section 6.1.: This whole section lacks a conclusion. First, which is the best estimate of the deviations, and second what could cause the strong deviations during the April flight (see also my major comment above). Have pollution/outgassing been considered to explain the differences?*

A more elaborate discussion has been provided. Outgassing was considered, and we have added this to the discussion.

*"Another potential source for the bias in the 26 April 2017 flight is outgassing from the packaging material and balloon. As the balloon ascents the surrounding pressure decreases and gasses desorb from the surface of the packaging material and balloon, which potentially influences our measurements. This would, however, not explain the good agreement during the September flights opposed to the April flights. Furthermore, the inlet is located at the top of the payload and any outgassing is flushed away from the inlet during ascent.”*

The suggestion of the influence NH seasonality, as in the major comment, has been accounted for.

*"The seasonality in tropospheric $CO_2$ could be another explanation to the observed difference. In northern hemisphere winter $CO_2$ mole fractions are higher, and hence the difference between sampled air and storage environment is higher. However, during the storage tests with low mole fractions, e.g. sample nos. 6&7 in Figure 2, the mole fractions of $CO_2$ were much lower than typical stratospheric samples. In that experiment the drift did not exceed 0.8 ppm. Hence, such a large drift cannot be fully attributed to the $CO_2$ variability in the storage environment that is due to seasonality.”*

A more detailed discussion on the uncertainties in AirCore has been added, see also the reply to the previous comment. We have no good reason to say that either one of the estimates is better, other than the faulty datalogger, therefore we conclude with deviations as follows:

*"Even though the AirCore might have been affected by diffusion and problems with the datalogger, AirCore and Sampler show good agreement in all flights, with mean AirCore sampler differences of 0.84 ppm for $CO_2$, 1.8 ppb for $CH_4$ and 6.3 ppb for CO.”*

*p. 13. L. 14.: this sentence is odd. How should a prolonged sampling time result in increased vertical resolution?*

The sentence has been rephrased. The explanation why prolonged sampling time compromises vertical resolution was presented in section 4.

*"Therefore, the gain in sample amount, for example adding 10 extra seconds of sampling time, is small; however, the decrease in vertical resolution is significant”*

*p. 14.l. 7.: this sentence is wrong, uncertainty is mentioned twice.*

The sentence has been corrected.

*"The uncertainty in sounding of the pressure sensors is 1 hPa at 200 hPa (Vaisala, 2013). The total uncertainty after 200 seconds of sampling is 9 mL$_{stp}$, slightly higher than the effect found above."*

*p. 14. Section 6.4. see major comment above. I believe that the contaminating fraction may be much higher.*

See our reply to the major comment.

*p. 15. L. 24: this is not up to 0.84 ppm. 0.84 ppm is actually an average deviation if all samples are considered.*

Correct, we have rephrased the sentence and now use "mean difference" instead of "up to".

*"...showing the mean differences between AirCore and LISA of ..."*

*Figure 2: the colours for the different bags types are virtually undistinguishable.*

The colours of all figures have changed. With respect to Figure 2, we have also increased the marker size to improve visibility.

*Figure 3: I wonder if all panels are needed here. I suggest removing panel b.*

Panel b shows the linear behaviour observed between sampled volume and ambient pressure. This allows us to model the volume sampled according to Eq. (2) where a(t) is then the only parameter depending on sampling time. To justify our model, we deem it necessary to show panel b. Panel c shows the fit that models the time dependence of the linear coefficient a(t) according to Eq. (3).

*Table 6: please explain the different columns on the table heading, not only in the text.*

We have provided more detailed explanations of the different columns in the revised version, now it is updated to Table 7:

*"Table 7: Expected bias in stable isotope measurements on samples obtained by LISA, due to the limited accuracy of the LISA sampler. Typical values for the troposphere and stratosphere are taken from the indicated references: A) (Trolier et al., 1996) B) (Mrozek et al., 2016) C) (Nisbet et al., 2016) D) (Bergamaschi et al., 2001) E) (Aoki et al., 2003) and F) (Röckmann et al., 2011). Reported measurement reproducibility's, Re, for stratospheric air are also provided. $\boldsymbol{\delta^{13}C}$ and $\boldsymbol{\delta^{18}O}$ values are with respect to Vienna Pee Dee Belemnite (VPDB) and $\boldsymbol{\Delta^{17}O}$ and $\boldsymbol{\delta^{2}H}$ are with respect to Vienna Standard Mean Ocean Water (VSMOW). $\boldsymbol{f_c}$ was calculated using a source value 395 ppm (CO$_2$) and 500 ppb (CH$_4$). For $\boldsymbol{f_{c1}}$ contamination values of 0.84 ppm (CO$_2$) and 1.8 ppb (CH$_4$) based on LISA AirCore observed mean bias, resulting in $\Delta\boldsymbol{\delta_1}$; For $\boldsymbol{f_{c2}}$ the maximum observed drift (Figure 2) of 0.11 ppm (CO$_2$) and 2 ppb (CH$_4$) are used, resulting in $\Delta\boldsymbol{\delta_2}$."*

References

Gerber, E. P., Butler, A., Calvo, N., Charlton-Perez, A., Giorgetta, M., Manzini, E., Perlwitz, J., Polvani, L. M., Sassi, F., Scaife, A. A., Shaw, T. A., Son, S. W. and Watanabe, S.: Assessing and understanding the impact of stratospheric dynamics and variability on the earth system, Bull. Am. Meteorol. Soc., 93(6), 845–859, doi:10.1175/BAMS-D-11-00145.1, 2012.

**Reply to Interactive comment of Referee #2 on "LISA: a lightweight stratospheric air sampler" by Joram J. D. Hooghiem et al.**

We thank the two reviewers for their helpful and detailed comments on our manuscript. We have addressed the major comments from both reviewers regarding our derivation of the bias estimation in isotopic composition measurements and revised the introduction according to the comments of both reviewers. Below, a detailed point-to-point reply and a revised version of our manuscript with track changes are given.

*General comments*

*The authors present a new stratospheric sampling system that can fill a niche between current techniques, as it provides larger air samples with better vertical resolution than AirCore while operating with much lower cost and payload weight than typical cryogenic whole air systems. The manuscript is suitable for AMT and well-written and should be published after addressing the following comments and requests. In particular, a possible error in the calculation of uncertainty for isotopic analysis should be resolved.*

*Major issue: Section 6.4*

*I agree with Reviewer 1 that the underlying assumptions used to estimate fraction of sample from contamination (fc) are incorrect. The ultimate source of the error seems to derive from the expression used to define fc =bias/([X] + bias). Here I presume that [X], defined by the authors as "typical mole fraction," is therefore meant to be [X] $\tilde{m}$, the measured mixing ratio.*

*If we rearrange equation (7) to δm = δs – Δδ and rewrite for mixing ratio as [X]m = [X]s – ΔX, where [X]s is the stratospheric mixing ratio and ΔX is the bias, then substitute this into the equation above, we have fc = ΔX / (ΔX – [X]s + ΔX) = ΔX / [X]s. But since the authors also define fc = ΔX / ([X]s – [X]c) in equation (8), this implies that the authors are assuming the mixing ratio of the contamination is 0 when they calculate the fc on pg 14, line 28. Instead, they should use measured or typical mixing ratios, nominal 400 ppm and 1800 ppb for CO2 and CH4, respectively. This would result in roughly the same estimate as Reviewer 1 has calculated.*

We would like to refer to the reply the comments of reviewer 1. We have copied it here for convenience.

We thank the reviewer for pointing out the calculation error. We mistakenly assumed that the contamination results from one-way diffusion of ambient $CO_2$ into the bag. If the observed bias would be entirely caused by diffused contamination air, the fraction of the contamination air is indeed about 8 percent. However, the observed bias is unlikely caused by diffusion alone based on our laboratory experiment results, where we found much smaller biases (~0.1 ppm for $CO_2$ and ~2 ppb for $CH_4$), as shown in Figure 2.

We have further thoughts on the diffusion process. According to the Fick's law, the diffusion depends on the diffusivity and the concentration gradient, which may be different for different species, e.g. the concentration gradient for $O_2$ and $N_2$ across the bag's material is negligible, and the concentration gradients for $CO_2$ and $CH_4$ depend on their ambient concentrations. We have updated the manuscript with the following analysis of the problem.

[revised manuscript text omitted]

*An alternative method to estimate fc would be to solve equation (8) for both CO2 and CH4 simultaneously. Since the authors already have estimates for $\Delta\delta$ and $\delta s$ stated (pg 14, lines 26-28), this results in two equations with two unknowns, which can then be solved trivially. This would be a worthwhile exercise to confirm the estimate of fc provided by Reviewer 1.*

Since the diffusion through the bag's material is species dependent (e.g. species dependent diffusivity and concentration gradient.), the fraction fc is also different for each species. That is why we find different values for $f_c$ as presented in table 7. In a simple mixing scheme with two air masses the suggested calculation would indeed be true.

*Finally, it is not clear to me why the authors don't use the storage test results, in addition to the ambient data, to estimate fc. This should either be performed, or the authors should explain why this analysis is not valid.*

In the revised manuscript we have added the estimate based on the storage test results.

*Additional general comments*

*I agree with Reviewer 1 that the introduction is a bit unfocused and broad. I think a more focused discussion of the literature with regards to the particular challenges of making sufficiently accurate and precise mixing ratio measurements in the stratosphere would be more useful. See the introduction offered by Membrive et al., 2017 for an example.*
*The manuscript presents multiple tables of information (e.g. Table 2 is derived fit coefficients, Table 3 is instrument operational settings) that would be better suited in supplemental materials, as this information is not critical to main discussion here). Meanwhile, there is no table provided that summarizes the instrument specifications (e.g. weight, power requirements, sample resolution at stated altitude) and comparison to the AirCore and other systems. I would be appreciative of such a summary table.*

We have added a table summarizing the key components of the sampler in section 2. We like to draw the attention to table 5 where a comparison of sample size and resolution between AirCore and Sampler, which to our best knowledge are to only to instruments with weight lower than 5 kg that can sample from the stratosphere.

In the light of the discussion presented in sections 6.2 about sampling size and vertical resolution we deemed the information in Table 3 relevant. Table 2 is removed from the manuscript, as we agree I does not provide any relevant information.

*"Table 1: Components used in the LISA sampler, including manufacturer and product key. The total weight is given for amounts per part. Voltage and power are presented according to manufacturer specification. The total weight for the onboard computer and sensors is given."*

| Component | Company | Product key | Amount | Voltage (V) | Power (W) | Weight (g) |
|---|---|---|---|---|---|---|
| Servo motor | Hitec | HS-65HB+ | 5 | 4.8-6 | 1.32 | 91 |
| Pump | KNF | NMP 850.1.2 KNDC B | 1 | 24 | 10.8 | 403.6 |
| Bag (MLF) | Supelco | 30227-U | 4 | (-) | (-) | 80.4 |
| Tube | Cole Palmer | EW-95100-02 | 1 | (-) | (-) | 30 |
| Union T | Swagelok | NY-400-3 | 5 | (-) | (-) | 39 |
| Union Knee | Swagelok | NY-400-9 | 5 | (-) | (-) | 33 |
| Battery | (-) | CR123 | 10 | 3 | (-) | 166 |
| Pressure sensor | Honeywell | HSCMAND015PASA5 | 2 | | | |
| Temperatrue sensor | IST | 600C (100Ohm) | 1 | 7-12 | (-) | 87.4 |
| Datalogger | Arduino | Mega 2560 | 1 | | | |
| Battery Casing | TruPower | BH-CR123A | 10 | (-) | (-) | 68.8 |

*Additional comments*

*p. 1, line 11 - The abstract discusses the stability tests, which include H2O mole fractions. However, the rest of the abstract only discuss CO2, CH4 and CO mole fractions. A phrase noting the significance of H2O (interferent?) would be appreciated.*

We have removed $H_2O$ from the abstract. $H_2O$ was measured and used to obtain dry mole fractions of $CO_2$, $CH_4$ and CO, as mentioned in Section 3.1. We are interested in the stability of $H_2O$ in the bags as well, due to its potential effect on isotopic compositions of CO2. $H_2O$ mole fraction measurements are now only left as a discussion in section 3.

*"H₂O measurements are used to obtain dry mole fraction of $CO_2$, $CH_4$ and CO using the water vapour corrections described in Chen et al., 2013 and Rella et al., 2013, before assessing drift of these species."*

*p. 1, line 29 – "Stratospheric changes in ozone and water vapour levels" is awkwardly phrased. How about "Changes in stratospheric ozone and water vapour levels"?*

We have adopted the suggested phrase.

*p. 2, line 1 – "stratospheric air up to 35 km" Please define this value (e.g. ASL).*

"Above mean sea level (a.m.s.l.)" has been added.

*p. 2, line 13 – "remarkable scientific efforts" I would avoid using words like "remarkable" without a substantial defence of this term.*

We have adjusted the sentence and avoided the term "remarkable" care has been taken to avoid such terms throughout the text.

*p. 2, line 15 – Introducing mean age of air is unnecessary here unless it is discussed later in the text.*

We have left out the discussion of mean age since it is indeed not relevant to this manuscript.

*p. 3, line 8 – "AirCore does not provide large sample amount" Please quantify this statement.*

We have added typical values of 300 to 600 mL of stratospheric air (200 to 0 hPa) sampled with AirCore.

*"The volume of air sampled between 0 to 200 hPa (12 to 30 km) by the AirCore ranges from 300 to 600 mL, depending on the geometry of the AirCore."*

p. 4, line 1 – "a diaphragm pump (KNF, product no. NMP 850.1.2 KNDC B)" Please provide a description of wetted surfaces for this pump.

A description has been added in the manuscript, after personal communication with KNF.

*"The pump utilizes an EPDM rubber diaphragm (35 mm diameter) and valves, and a small piece of flexible PU tube."*

*p. 5, line 20 – Storage test results. Can the authors take advantage of the results of these tests to separate leak and permeability effects for all species?*

Diffusion through the bags material is species dependent. Furthermore, the diffusivity through the materials is unknown, and determination of the diffusivities require experiments beyond the scope of this work. Therefore, the experiments do not allow to separate leak and permeability effects. Besides these, most of the time the pressure difference between in and outside the bag is 0, hence there is no driving force for air to leak in or out. Only during ascent, the pressure inside the bag is higher than ambient and air might leak out. Thus, diffusion

through the material and possibly through the septum of the inlet is likely the main cause of the observed drift in mole fractions during the storage tests.

*The discussion of the water vapor experiment is nicely done and touches on observed changes in CO; can this be extended?*

With our current knowledge and data, we can only speculate the different reasons for the observed changes in CO. In the manuscript the only remaining correlation we could find from the data is that with water vapour. At this stage we can only postulate a potential bias in the water vapour correction function. Though the water vapour corrections are well quantified, they are not tested with low mole fractions.

*p. 6, line 11 – "we observed an offset" Please describe which direction these offsets were (increase or decrease). As a reader, I wondered if this effect could have been a cause of the CO2 offset observed during the April flight discussed later.*

In the revised manuscript the direction, which is positive, has been clarified. There is no difference in preparation between the April and September flights. Due to unavailability of N2 the bags were preconditioned with our calibration gas in the field prior to each flight. If the offsets would be the cause, it should have been most notable in CO, with values higher than AirCore, which is not the case. We therefor conclude that the offsets prior to preconditioning contribute little to the observed AirCore-LISA discrepancies.

*"We found out that it is necessary to precondition the MLF bags before use, because we observed a positive offset of ~12 ppm CO2, ~8 ppb CH4 and ~30 ppb CO between the immediately analysed results of un-preconditioned MLF bag samples after filling and the assigned cylinder values"*

*p. 6, line 24 – "We do not include any sampling error in the presented evaluation, although it might be significant for high-precision measurements, especially for CO in the stratosphere when the ozone concentration is high." This statement was ambiguous to me. Please explain exactly what is meant by "sampling error" and the significance of ozone concentration.*

At the time of writing, we hypothesized that ozone might be reacting with the pump materials. However, this was checked later and apparently ozone concentrations of up to 1000 ppm does not have a significant effect on the EPDM material: http://www.ozoneapplications.com/info/ozone_compatible_materials.htm. The sentence has thus been removed.

The sampling error encompasses any contamination effects or reactions that might alter the mole fraction during the sampling process. This has been adopted in the revised manuscript.

*"The sampling error encompasses any contamination introduced by the sampling system itself. This includes chemical production of the species of interest and residual air in any dead volumes of the manifold. The chemical production during sampling is likely to be very small for two reasons. First of all, the wetted surfaces, Kynar and EPDM diaphragm, are chemically inert. Secondly, the high flow rate minimizes exposure of the sample to materials used in the sampler and hence chemical interaction with the wetted surfaces is limited. In addition, the flushing procedure with high flowrates ensures multiple turnovers of the manifold, which*

*reduces the surface effects on the sample. These effects are thus assumed to have no influence on the CO$_2$, CH$_4$ and CO mole fractions.*

*The dead volumes in the manifold are a potential source of contamination bias. The dead volume is estimated to be 1.5 mL per sample and will be at local ambient pressure prior to sampling. So, the dead volume uncertainty, $\sigma_v$, to the contribution to a sample at 200 L$_{stp}$ sampling is very small."*

*p. 6, line 28 – "However, detailed information about storage conditions are required to correct for the drift, which is usually unavailable in the field." Why wouldn't this be available? The detector used for analysis was in the field (see p. 9, line 27) so this data may have been collected for the samples presented here. Please explain if the data is available and whether it should be collected in future work.*

The statement in P. 9, Line 27 (original manuscript) was ambiguous. The samples were analysed in the FMI laboratory at the Sodankylä observatory, which to us, being from Groningen, is "in the field". The statement has now been modified in the revised manuscript.

*"After retrieval of the payload, the samples were analysed in the TCCON laboratory"*

The detailed storage information includes 1) ambient mole fractions of different species; 2) ambient temperature; 3) ambient pressure. The ambient temperature and pressure were logged by the datalogger, and the ambient mole fraction measurements of different species were not done as it would require a dedicated precise and accurate analyser. Since a robust correlation between the ambient information and the drift during storage has not been established, we would not recommend collecting this information in future work.

*p. 7, line 12 – "The performance of the small diaphragm pump is to our best knowledge not previously investigated under the atmospheric conditions in the stratosphere, e.g. at low-temperature and low-pressure conditions." This is a nice presentation of this experiment. Was the pump subject to stratospheric temperatures? The manufacturers specification is for ambient temperature between +5˚C and +40˚C. Any insight that can be provided about the pump performance at stratospheric temperatures would be appreciated.*

Indeed, no laboratory tests were performed at stratospheric temperatures. The pump performance could be affected by the cold environment. First, the batteries could lose capacity and cause the power supplied to the pump decreasing. The temperature inside the thermally insulated package, where the pump is located, during flight ranged between 30 and -15 ˚C. Secondly, the diaphragm is exposed to the cold air passing through the pump. The elasticity of a rubber is temperature dependent, which could reduce the performance of the pump. On the other hand, heat is released during operation of the pump, which increases the temperature. Finally, the effect of air temperature on sample size follows the ideal gas law, and the sample size increases at low temperatures. As no experimental data is available to determine the performance of the pump at stratospheric temperature, we assume that the pump performs the same during flight as at room temperature for the calculation of the sample size. The best insight we have is the comparison of the flight results in Fig. 4.

We have clarified in the manuscript with the following text in section 6.

*"During the experiments described in Section 4, the pump was at room temperature. The pump performance could be affected by the cold environment. First, the batteries could lose capacity and cause the power supplied to the pump to decrease. The temperature inside the thermally insulated package, where the pump is located, during flight ranged between 30 and -15 ˚C. Secondly, the diaphragm is exposed to the cold air passing through the pump. The elasticity of a rubber is temperature dependent, which could reduce the performance of the pump. On the other hand, heat is released during operation of the pump, which increases the temperature. Finally, the effect of air temperature on sample size follows the ideal gas law, and the sample size increases at low temperatures. As no experimental data is available to determine the performance of the pump at stratospheric temperature, we assume that the pump performs the same during flight as at room temperature for the calculation of the sample size."*

*p. 8, line 6 – "19.7 seconds." How precise is this value? I'm a bit surprised that the pumping speed would be so consistent at varying pressure. There are no data points shown in Figure 3 prior to 20 sec, which makes this statement a bit hard to evaluate. The uncertainty of this value under operating conditions should be discussed with respect to the modelled behaviour that follows.*

Initially pressure is constant as the bag is expanding. When expanded to its full size, the pressure starts increasing, which was found to be around 19.7 seconds which gave the best fit. This was based on visual inspection of the data. We have added an estimate of the accuracy of this number of about plus or minus 1 second, since we cannot justify otherwise.

*"The sampled air volume increases linearly with the sampling time when the bag expands to its full size during the first 20±1 second's. The moment compression is required, pressure starts increasing rapidly, and this moment was found to be 20 seconds after filling initiated."*

*p. 8, line 25 – "The sampled volume at STP can be modelled for all pressure levels ranging from 200 to 0 hPa" It seems inappropriate to model to 0 hPa, given that the data collected ends at 30 hPa.*

We have rephrased "modelled" to "approximated". This was ultimately the goal of the experiment. It provides us with a tool to determine the sampling strategy in the field. And the approximation suffices.

*"Combining Eq. (1) and Eq. (2), the sampled volume at STP can be approximated for all pressure levels ranging from 200 to 0 hPa for any chosen sampling time."*

*p. 9, line 5 – "To be on the safe side," should be rephrased "Conservatively,"*

The sentence has been rephrased according to the suggestion made by the reviewer.

*"Conservatively, we set the maximum absolute pressure in the MLF bag during flight not higher than 280 hPa to avoid any potential loss of sample due to the burst of the bag."*

p. 11, line 10 – "the vertical resolution increases" The vertical resolution decreases since the vertical height increases. There are several significant deviations from the relationship between altitude and vertical resolution, as shown in Figure 4b: 26-Apr at 150 hPa, 4-Sep at 100 hPa and 5-Sep at 30 hPa. Could these data points be explained in the context of the discussion of the variability in vertical resolution starting at p. 11, line 14?

Indeed, the vertical resolution decreases with altitude. As explained in P. 11, Line 28 there were 3 factors that explained all the existing variability 1) varying sampling time; 2) varying ascending speed; 3) varying maximum allowed pressure.
We have highlighted the samples mentioned by the reviewer above in the manuscript.

*"The variability in the vertical resolution is the result of three factors: 1) varying sampling time; 2) varying ascending speed; 3) varying maximum allowed pressure. The ascending speed was typically around 7-9 m s$^{-1}$ in the lower stratosphere and decreased to 4-5 m s$^{-1}$ in the middle stratosphere. The varying ascending speed accounts for the observed deviations from the otherwise linear trend in Fig. 4b. In the lower stratosphere (10 to 15) km the maximum allowed pressure inside the bags was usually reached in a period shorter than the pre-set sampling time, leading to relatively high vertical resolution. In the region 10 to 15 km two samples deviate (4-Sep 17 km and 26-April 14 km), with lower resolution, which is due to a higher ascending speed. In the middle stratosphere, the sampling time was usually the limiting factor to vertical resolution. One sample in the middle stratosphere has a relatively good vertical resolution (5-Sep, 25 km), which is due to the relatively slow ascent speed."*

*p. 11, line 25 – While the vertical profiles of the mixing ratios are shown in Figure 5, I don't see the actual values presented. Could this be added to Table 3 or as a Supplemental Table (along with individual uncertainties of both mixing ratio and vertical height)?*

The data is available on request.

*p. 12, line 8 - While there are significant differences between the LISA and AirCore retrieved mixing ratios for CO2, the CH4 and CO mixing ratios show good agreement. Can this allow the authors to make a statement about the validity of their assumptions for estimating altitudes and pressures for the failed data loggers? And does this indicate that the offset CO2 measurement is a result of the measurement of mixing ratio rather than altitude? This comment is also for p. 12 line 30 to p. 13, line 2.*

This is a very good suggestion, which we have added in our discussion (section 6.1). The CO is not a good diagnostic since virtually no gradient is observed in CO (within measurement uncertainty). CH$_4$ however is a good diagnostic and indeed the observed LISA and AirCore CH$_4$ profiles show that the altitude registration is reliable. The CO$_2$ offset must be related to a difference in measurement and cannot be attributed fully to altitude registration.

*"The AirCore valve did not close during the 26 April 2017 flight. This complicates the altitude registration of the AirCore. However, the large degree of agreement in CH$_4$ profiles between AirCore and LISA shows that AirCore altitude registration is reliable for the flight on 26 April 2017. The bias in CO$_2$ cannot be attributed to a failure in altitude registration and hence must be related to mole fraction measurement."*

*p. 12, line 17 – "0.5 ppm or 0.13%" I'm not sure that the percentage reported here is appropriate, as it implies greater analytical precision that is achieved. If the CO2 mixing ratio varies only 20 ppm or less between the troposphere and the top of the sampling region, then 0.5 ppm is a more significant deviation. I would suggest simply removing these percentages.*

We agree that depiction of a percentage leaves room for interpretation. The percentages are thus removed from the manuscript.

*p. 13, line 13 – "decreases fast." Should be "decreases quickly." In the following sentence, "the gain in sample amount, for example adding 10 extra seconds of sampling time, does not increase" is not correct, but rather the gain in sample amount is very small in comparison to the decrease in vertical resolution. This should be restated.*

The sentence in the original manuscript was indeed ill phrased and has been updated according to the reviewer's comment.

*"From the results shown in Fig. 3d, we see that after 200 seconds of sampling the gain in sample amount decreases quickly. Therefore, the gain in sample amount, for example adding 10 extra seconds of sampling time, is small, however the decrease in vertical resolution is significant."*

*p. 13, line 16 - As an aside, I would suggest additional bags as an alternative to larger bags, as the altitude profiles presented in Figure 5 indicate that there are periods during the ascent when no sample is collected, i.e. the system is sitting idle.*

We agree that additional bags would be a good option to increase the vertical density of the sampling, and the system presented in this manuscript can indeed be easily adapted for that. However, for other scientific needs, e.g. for more precise 14CO2 measurements, larger samples are preferred. We have modified the text to include both cases.

*"Alternatively, to increase the amount of sample retrieved during one flight, additional bags can be considered. Currently the system is idle during several stages of the ascent as can be inferred from Figure 5. This will however be more demanding on battery power. Furthermore, care has to be taken to avoid overlapping sampling schemes i.e. sampling of a sample at altitude $P_1$ is still ongoing while the set-point altitude for sample two, $P_2$ is reached. This is complicated further with variable ascent speed, that is typical for these balloon flights."*

*p. 13, line 28 – It is not clear why the uncertainty in the sample amount is important. Could the authors explain the importance of this?*

The sample amount is important because it often significantly influences the precision of the measurements by instruments. Since we aim to obtain larger sample amount and the sample amount was only determined indirectly from pressure and temperature measurements, we think it logical to estimate the uncertainty of the sample size. We have added the uncertainty to the captions of Figures 3 - 4.

*p. 23 – Figure 3b. I found the reuse of marker shapes between 3a and 3b confusing (I was looking for a relationship between the same marker shape in both figures). Could the authors change the marker shapes in Figure 3b to avoid this? For Fig 3c, it is not clear if the units on the vertical axis use volumetric liters or standard liters. This should be standard liters for consistency. Also, in Fig 3d please extend the vertical line at 0.76 L for the full vertical height of the figure, as was done in Fig 3b.*

We have adjusted the marker shapes so that 3a and 3b are now compatible. The vertical line is extended.

*p. 25 – Figure 4 and Figure 5. As a color-blind reader I cannot distinguish between the colors used for the 26-Apr and 4-Sep flight data. Could the 26-Apr color be changed to a different color (e.g. blue)?*

Yes, we have updated the figures with colour blind friendly palettes, and tested them with the following simulator:

http://www.color-blindness.com/coblis-color-blindness-simulator/

*p. 27 – Table 6. Please define the acronyms VPDB and VSMOW.*

The definitions have been added in the revised version, and now it is table 7.

Reference: Membrive, O., Crevoisier, C., Sweeney, C., Danis, F., Hertzog, A., Engel, A., Bönisch, H., and Picon, L.: AirCore-HR: a high-resolution column sampling to en- hance the vertical description of CH4 and CO2, Atmos. Meas. Tech., 10, 2163-2181, https://doi.org/10.5194/amt-10-2163-2017, 2017.

[revised manuscript text omitted]

Formatted ... [43]

Formatted ... [44]

Formatted ... [45]

Formatted ... [46]

Formatted ... [47]

Formatted ... [48]

Formatted ... [42]

Formatted ... [49]

Formatted ... [50]

Formatted ... [51]

Formatted ... [52]

Formatted ... [53]

¶

| Page 2: [1] Deleted | Joram Hooghiem | 4/5/18 3:24:00 PM |

Nevertheless, a significant number of stratospheric air samples have been collected with remarkable scientific efforts. Several campaigns have been performed and corresponding results have been used in many studies of the stratospheric chemistry and physics. Laube et al., 2010, presented several profiles of halocarbons, relevant to stratospheric ozone depletion and the mean age of air. The mean age of air, a good tracer for atmospheric transport time scales, was also assessed based on $SF_6$ and $CO_2$ measurements performed on cryogenically retrieved samples (Engel, 2002). Long-term monitoring of $CO_2$, $CH_4$, $N_2O$ and various halocarbons and isotopic analysis of $CO_2$, $CH_4$, $N_2O$ have been performed annually for more than a decade (Aoki et al., 2003; Nakazawa et al., 1995, 2002). The stratospheric distribution of methane and its stable isotopes have been performed in order to understand the stratospheric methane sink (Rice, 2003; Röckmann et al., 2011; Sugawara et al., 1997). The stratospheric distribution of $N_2O$ and their position dependent isotopic compositions were determined (Kaiser et al., 2006) and were subsequently used to validate transport models. Engel et al., 2006, reported the observed mesospheric air in polar vortex, improving our understanding of transport in the middle atmosphere. Moreover

| Page 2: [2] Deleted | Joram Hooghiem | 4/5/18 3:25:00 PM |

, these campaigns are essential to validation and comparison of satellite retrievals (e.g. Engel et al., 2016; Stiller et al., 2007).

Trace gas distributions provide a useful tool to obtain an insight in transport properties and chemistry of the stratosphere. (Brenninkmeijer et al., 1995) studied the correlation of $CO_2$, $CH_4$ and $CO$ mole fractions and their isotopic composition measurements, and found a high correlation between mole fractions and isotopic compositions. It was shown that the $^{13}C$ and $^2H$ isotopic composition of stratospheric $CH_4$ is strongly correlated with its mole fractions (Röckmann et al., 2011) and similar findings were presented for $N_2O$ (Kaiser et al., 2006). A theoretical explanation to the tracer-tracer relations due to rapid mixing of air along isentropic surfaces in the stratosphere was presented by (Plumb, 2007).

Stratospheric tracer observations are essential for validation of General Circulation Models (GCM's). The stratospheric meridional overturning, or the Brewer-Dobson circulation (BDC) was predicted to increase in strength from modelling studies (Butchart, 2014). The mean age of stratospheric air samples was shown to be a good diagnostic for the strength of the BDC and detected no significant change in the strength of the BDC (Engel et al., 2009, 2017). In spite of all the efforts to make observations of stratospheric tracers, GCM's remain poorly constrained, a problem already pointed out several decades ago (Ehhalt et al., 1983).

| Page 16: [3] Formatted | Joram Hooghiem | 7/13/18 4:42:00 PM |

Underline, English (UK)

| Page 16: [4] Formatted | Joram Hooghiem | 7/13/18 4:42:00 PM |

Underline, English (UK)

| Page 16: [4] Formatted | Joram Hooghiem | 7/13/18 4:42:00 PM |

Underline, English (UK)

| Page 16: [4] Formatted | Joram Hooghiem | 7/13/18 4:42:00 PM |

Underline, English (UK)

| Page 16: [5] Formatted | Joram Hooghiem | 7/13/18 4:42:00 PM |

Underline, English (UK)

| Page 16: [5] Formatted | Joram Hooghiem | 7/13/18 4:42:00 PM |

Underline, English (UK)

| Page 16: [5] Formatted | Joram Hooghiem | 7/13/18 4:42:00 PM |

Underline, English (UK)

| Page 16: [6] Formatted | Joram Hooghiem | 7/13/18 4:42:00 PM |

Underline, English (UK)

| Page 16: [6] Formatted | Joram Hooghiem | 7/13/18 4:42:00 PM |

Underline, English (UK)

| Page 16: [6] Formatted | Joram Hooghiem | 7/13/18 4:42:00 PM |

Underline, English (UK)

| Page 16: [6] Formatted | Joram Hooghiem | 7/13/18 4:42:00 PM |

Underline, English (UK)

| Page 16: [6] Formatted | Joram Hooghiem | 7/13/18 4:42:00 PM |

Underline, English (UK)

| Page 16: [7] Formatted | Joram Hooghiem | 7/13/18 4:42:00 PM |

Underline, English (UK)

| Page 16: [7] Formatted | Joram Hooghiem | 7/13/18 4:42:00 PM |

Underline, English (UK)

| Page 16: [7] Formatted | Joram Hooghiem | 7/13/18 4:42:00 PM |

Underline, English (UK)

| Page 16: [7] Formatted | Joram Hooghiem | 7/13/18 4:42:00 PM |

Underline, English (UK)

| Page 16: [7] Formatted | Joram Hooghiem | 7/13/18 4:42:00 PM |

Underline, English (UK)

| Page 16: [7] Formatted | Joram Hooghiem | 7/13/18 4:42:00 PM |

Underline, English (UK)

| Page 16: [8] Formatted | Joram Hooghiem | 7/13/18 4:42:00 PM |

Underline, English (UK)

| Page 16: [8] Formatted | Joram Hooghiem | 7/13/18 4:42:00 PM |
|---|---|---|

Underline, English (UK)

| Page 16: [8] Formatted | Joram Hooghiem | 7/13/18 4:42:00 PM |
|---|---|---|

Underline, English (UK)

| Page 16: [9] Formatted | Joram Hooghiem | 7/13/18 4:42:00 PM |
|---|---|---|

Underline, English (UK)

| Page 16: [9] Formatted | Joram Hooghiem | 7/13/18 4:42:00 PM |
|---|---|---|

Underline, English (UK)

| Page 16: [9] Formatted | Joram Hooghiem | 7/13/18 4:42:00 PM |
|---|---|---|

Underline, English (UK)

| Page 16: [10] Formatted | Joram Hooghiem | 7/13/18 4:42:00 PM |
|---|---|---|

Underline, English (UK)

| Page 16: [10] Formatted | Joram Hooghiem | 7/13/18 4:42:00 PM |
|---|---|---|

Underline, English (UK)

| Page 16: [10] Formatted | Joram Hooghiem | 7/13/18 4:42:00 PM |
|---|---|---|

Underline, English (UK)

| Page 16: [10] Formatted | Joram Hooghiem | 7/13/18 4:42:00 PM |
|---|---|---|

Underline, English (UK)

| Page 16: [10] Formatted | Joram Hooghiem | 7/13/18 4:42:00 PM |
|---|---|---|

Underline, English (UK)

| Page 16: [11] Formatted | Joram Hooghiem | 7/13/18 4:42:00 PM |
|---|---|---|

Underline, English (UK)

| Page 16: [11] Formatted | Joram Hooghiem | 7/13/18 4:42:00 PM |
|---|---|---|

Underline, English (UK)

| Page 16: [11] Formatted | Joram Hooghiem | 7/13/18 4:42:00 PM |
|---|---|---|

Underline, English (UK)

| Page 16: [12] Formatted | Joram Hooghiem | 7/13/18 4:42:00 PM |
|---|---|---|

Underline, English (UK)

| Page 16: [12] Formatted | Joram Hooghiem | 7/13/18 4:42:00 PM |
|---|---|---|

Underline, English (UK)

| Page 16: [12] Formatted | Joram Hooghiem | 7/13/18 4:42:00 PM |
|---|---|---|

Underline, English (UK)

| Page 16: [13] Formatted | Joram Hooghiem | 7/13/18 4:42:00 PM |
|---|---|---|

Underline, English (UK)

| Page 16: [13] Formatted | Joram Hooghiem | 7/13/18 4:42:00 PM |
|---|---|---|

Underline, English (UK)

| Page 16: [13] Formatted | Joram Hooghiem | 7/13/18 4:42:00 PM |
|---|---|---|

Underline, English (UK)

| Page 16: [14] Formatted | Joram Hooghiem | 7/13/18 4:42:00 PM |
|---|---|---|

Underline, English (UK)

| Page 16: [14] Formatted | Joram Hooghiem | 7/13/18 4:42:00 PM |
|---|---|---|

Underline, English (UK)

| Page 16: [14] Formatted | Joram Hooghiem | 7/13/18 4:42:00 PM |
|---|---|---|

Underline, English (UK)

| Page 16: [15] Formatted | Joram Hooghiem | 7/13/18 4:42:00 PM |
|---|---|---|

Underline, English (UK)

| Page 16: [15] Formatted | Joram Hooghiem | 7/13/18 4:42:00 PM |
|---|---|---|

Underline, English (UK)

| Page 16: [15] Formatted | Joram Hooghiem | 7/13/18 4:42:00 PM |
|---|---|---|

Underline, English (UK)

| Page 16: [16] Formatted | Joram Hooghiem | 7/13/18 4:42:00 PM |
|---|---|---|

Underline, English (UK)

| Page 16: [16] Formatted | Joram Hooghiem | 7/13/18 4:42:00 PM |
|---|---|---|

Underline, English (UK)

| Page 16: [16] Formatted | Joram Hooghiem | 7/13/18 4:42:00 PM |
|---|---|---|

Underline, English (UK)

| Page 16: [17] Formatted | Joram Hooghiem | 7/13/18 4:42:00 PM |
|---|---|---|

Underline, English (UK)

| Page 16: [17] Formatted | Joram Hooghiem | 7/13/18 4:42:00 PM |
|---|---|---|

Underline, English (UK)

| Page 16: [17] Formatted | Joram Hooghiem | 7/13/18 4:42:00 PM |
|---|---|---|

Underline, English (UK)

| Page 16: [17] Formatted | Joram Hooghiem | 7/13/18 4:42:00 PM |
|---|---|---|

Underline, English (UK)

| Page 16: [17] Formatted | Joram Hooghiem | 7/13/18 4:42:00 PM |
|---|---|---|

Underline, English (UK)

| Page 16: [17] Formatted | Joram Hooghiem | 7/13/18 4:42:00 PM |
|---|---|---|

Underline, English (UK)

| Page 16: [17] Formatted | Joram Hooghiem | 7/13/18 4:42:00 PM |
|---|---|---|

Underline, English (UK)

| Page 16: [17] Formatted | Joram Hooghiem | 7/13/18 4:42:00 PM |
|---|---|---|

Underline, English (UK)

| Page 16: [17] Formatted | Joram Hooghiem | 7/13/18 4:42:00 PM |
|---|---|---|

Underline, English (UK)

| Page 16: [17] Formatted | Joram Hooghiem | 7/13/18 4:42:00 PM |
|---|---|---|

Underline, English (UK)

| Page 16: [18] Formatted | Joram Hooghiem | 7/13/18 4:42:00 PM |
|---|---|---|

Underline, English (UK)

| Page 16: [18] Formatted | Joram Hooghiem | 7/13/18 4:42:00 PM |
|---|---|---|

Underline, English (UK)

| Page 16: [18] Formatted | Joram Hooghiem | 7/13/18 4:42:00 PM |
|---|---|---|

Underline, English (UK)

| Page 16: [18] Formatted | Joram Hooghiem | 7/13/18 4:42:00 PM |
|---|---|---|

Underline, English (UK)

| Page 16: [18] Formatted | Joram Hooghiem | 7/13/18 4:42:00 PM |
|---|---|---|

Underline, English (UK)

| Page 16: [18] Formatted | Joram Hooghiem | 7/13/18 4:42:00 PM |
|---|---|---|

Underline, English (UK)

| Page 16: [18] Formatted | Joram Hooghiem | 7/13/18 4:42:00 PM |
|---|---|---|

Underline, English (UK)

| Page 16: [18] Formatted | Joram Hooghiem | 7/13/18 4:42:00 PM |
|---|---|---|

Underline, English (UK)

| Page 16: [18] Formatted | Joram Hooghiem | 7/13/18 4:42:00 PM |
|---|---|---|

Underline, English (UK)

| Page 16: [18] Formatted | Joram Hooghiem | 7/13/18 4:42:00 PM |
|---|---|---|

Underline, English (UK)

| Page 16: [18] Formatted | Joram Hooghiem | 7/13/18 4:42:00 PM |
|---|---|---|

Underline, English (UK)

| Page 16: [18] Formatted | Joram Hooghiem | 7/13/18 4:42:00 PM |
|---|---|---|

Underline, English (UK)

| Page 16: [18] Formatted | Joram Hooghiem | 7/13/18 4:42:00 PM |
|---|---|---|

Underline, English (UK)

| Page 16: [18] Formatted | Joram Hooghiem | 7/13/18 4:42:00 PM |
|---|---|---|

Underline, English (UK)

| Page 16: [18] Formatted | Joram Hooghiem | 7/13/18 4:42:00 PM |
|---|---|---|

Underline, English (UK)

| Page 16: [18] Formatted | Joram Hooghiem | 7/13/18 4:42:00 PM |
|---|---|---|

Underline, English (UK)

| Page 16: [18] Formatted | Joram Hooghiem | 7/13/18 4:42:00 PM |
|---|---|---|

Underline, English (UK)

| Page 16: [18] Formatted | Joram Hooghiem | 7/13/18 4:42:00 PM |
|---|---|---|

Underline, English (UK)

| Page 16: [18] Formatted | Joram Hooghiem | 7/13/18 4:42:00 PM |
|---|---|---|

Underline, English (UK)

| Page 16: [19] Formatted | Joram Hooghiem | 7/13/18 4:42:00 PM |
|---|---|---|

Underline, English (UK)

| Page 16: [19] Formatted | Joram Hooghiem | 7/13/18 4:42:00 PM |
|---|---|---|

Underline, English (UK)

| Page 16: [19] Formatted | Joram Hooghiem | 7/13/18 4:42:00 PM |
|---|---|---|

Underline, English (UK)

| Page 16: [20] Formatted | Joram Hooghiem | 7/13/18 4:42:00 PM |
|---|---|---|

Underline, English (UK)

| Page 16: [20] Formatted | Joram Hooghiem | 7/13/18 4:42:00 PM |
|---|---|---|

Underline, English (UK)

| Page 16: [20] Formatted | Joram Hooghiem | 7/13/18 4:42:00 PM |
|---|---|---|

Underline, English (UK)

| Page 16: [20] Formatted | Joram Hooghiem | 7/13/18 4:42:00 PM |
|---|---|---|

Underline, English (UK)

| Page 16: [20] Formatted | Joram Hooghiem | 7/13/18 4:42:00 PM |
|---|---|---|

Underline, English (UK)

| Page 16: [20] Formatted | Joram Hooghiem | 7/13/18 4:42:00 PM |
|---|---|---|

Underline, English (UK)

| Page 16: [21] Formatted | Joram Hooghiem | 7/13/18 4:42:00 PM |
|---|---|---|

Underline, English (UK)

| Page 16: [21] Formatted | Joram Hooghiem | 7/13/18 4:42:00 PM |
|---|---|---|

Underline, English (UK)

| Page 16: [21] Formatted | Joram Hooghiem | 7/13/18 4:42:00 PM |
|---|---|---|

Underline, English (UK)

| Page 16: [21] Formatted | Joram Hooghiem | 7/13/18 4:42:00 PM |
|---|---|---|

Underline, English (UK)

| Page 16: [21] Formatted | Joram Hooghiem | 7/13/18 4:42:00 PM |
|---|---|---|

Underline, English (UK)

| Page 16: [21] Formatted | Joram Hooghiem | 7/13/18 4:42:00 PM |
|---|---|---|

Underline, English (UK)

| Page 16: [22] Formatted | Joram Hooghiem | 7/13/18 4:42:00 PM |
|---|---|---|

Underline, English (UK)

| Page 16: [23] Formatted | Joram Hooghiem | 7/13/18 4:42:00 PM |
|---|---|---|

Underline, English (UK)

| Page 16: [24] Formatted | Joram Hooghiem | 7/13/18 4:42:00 PM |
|---|---|---|

Underline, English (UK)

| Page 16: [25] Formatted | Joram Hooghiem | 7/13/18 4:42:00 PM |
|---|---|---|

Underline, English (UK)

| Page 16: [26] Formatted | Joram Hooghiem | 7/13/18 4:42:00 PM |
|---|---|---|

Underline, English (UK)

| Page 16: [26] Formatted | Joram Hooghiem | 7/13/18 4:42:00 PM |
|---|---|---|

Underline, English (UK)

| Page 16: [26] Formatted | Joram Hooghiem | 7/13/18 4:42:00 PM |
|---|---|---|

Underline, English (UK)

| Page 16: [27] Formatted | Joram Hooghiem | 7/13/18 4:42:00 PM |
|---|---|---|

Underline, English (UK)

| Page 16: [27] Formatted | Joram Hooghiem | 7/13/18 4:42:00 PM |
|---|---|---|

Underline, English (UK)

| Page 16: [27] Formatted | Joram Hooghiem | 7/13/18 4:42:00 PM |
|---|---|---|

Underline, English (UK)

| Page 16: [27] Formatted | Joram Hooghiem | 7/13/18 4:42:00 PM |
|---|---|---|

Underline, English (UK)

| Page 16: [27] Formatted | Joram Hooghiem | 7/13/18 4:42:00 PM |
|---|---|---|

Underline, English (UK)

| Page 16: [27] Formatted | Joram Hooghiem | 7/13/18 4:42:00 PM |
|---|---|---|

Underline, English (UK)

| Page 16: [28] Formatted | Joram Hooghiem | 7/13/18 4:42:00 PM |
|---|---|---|

Underline, English (UK)

| Page 16: [28] Formatted | Joram Hooghiem | 7/13/18 4:42:00 PM |
|---|---|---|

Underline, English (UK)

| Page 16: [28] Formatted | Joram Hooghiem | 7/13/18 4:42:00 PM |
|---|---|---|

Underline, English (UK)

| Page 16: [28] Formatted | Joram Hooghiem | 7/13/18 4:42:00 PM |
|---|---|---|

Underline, English (UK)

| Page 16: [28] Formatted | Joram Hooghiem | 7/13/18 4:42:00 PM |
|---|---|---|

Underline, English (UK)

| Page 16: [28] Formatted | Joram Hooghiem | 7/13/18 4:42:00 PM |
|---|---|---|

Underline, English (UK)

| Page 16: [28] Formatted | Joram Hooghiem | 7/13/18 4:42:00 PM |
|---|---|---|

Underline, English (UK)

| Page 16: [28] Formatted | Joram Hooghiem | 7/13/18 4:42:00 PM |
|---|---|---|

Underline, English (UK)

| Page 16: [29] Formatted | Joram Hooghiem | 7/13/18 4:42:00 PM |
|---|---|---|

Underline, English (UK)

| Page 16: [29] Formatted | Joram Hooghiem | 7/13/18 4:42:00 PM |
|---|---|---|

Underline, English (UK)

| Page 16: [29] Formatted | Joram Hooghiem | 7/13/18 4:42:00 PM |
|---|---|---|

Underline, English (UK)

| Page 16: [30] Formatted | Joram Hooghiem | 7/13/18 4:42:00 PM |
|---|---|---|

Underline, English (UK)

| Page 16: [30] Formatted | Joram Hooghiem | 7/13/18 4:42:00 PM |
|---|---|---|

Underline, English (UK)

| Page 16: [30] Formatted | Joram Hooghiem | 7/13/18 4:42:00 PM |
|---|---|---|

Underline, English (UK)

| Page 16: [31] Formatted | Joram Hooghiem | 7/13/18 4:42:00 PM |
|---|---|---|

Underline, English (UK)

| Page 16: [31] Formatted | Joram Hooghiem | 7/13/18 4:42:00 PM |
|---|---|---|

Underline, English (UK)

| Page 16: [31] Formatted | Joram Hooghiem | 7/13/18 4:42:00 PM |
|---|---|---|

Underline, English (UK)

| Page 16: [32] Formatted | Joram Hooghiem | 7/13/18 4:42:00 PM |
|---|---|---|

Underline, English (UK)

| Page 16: [32] Formatted | Joram Hooghiem | 7/13/18 4:42:00 PM |
|---|---|---|

Underline, English (UK)

| Page 16: [32] Formatted | Joram Hooghiem | 7/13/18 4:42:00 PM |
|---|---|---|

Underline, English (UK)

| Page 27: [33] Deleted | Joram Hooghiem | 4/5/18 4:17:00 PM |
|---|---|---|

Table 2: Fit constants for Eq. (3) and standard error of the fit.

| Parameter | Value | Error (1 $\sigma$) |
|---|---|---|
| x | 0.0080 | 0.00003 |
| b | 0.0050 | 0.00002 |
| $\tau$ | 59.61 | 0.82 |

| Page 28: [34] Deleted | Joram Hooghiem | 3/23/18 9:27:00 AM |
|---|---|---|

[Figure]

| | |
|---|---|
| ◆ Sampler 26-04-2017 | ★ Sampler 05-09-2017 |
| ● Sampler 04-09-2017 | ▶ Sampler 06-09-2017 |

**Page 28: [34] Deleted**        Joram Hooghiem        3/23/18 9:27:00 AM

[Figure]

[Figure]

| | | | |
|---|---|---|---|
| ◆ | Sampler 26-04-2017 | ★ | Sampler 05-09-2017 |
| ● | Sampler 04-09-2017 | ▷ | Sampler 06-09-2017 |

Page 28: [34] Deleted       Joram Hooghiem       3/23/18 9:27:00 AM

[Figure]

| Page 28: [34] Deleted | Joram Hooghiem | 3/23/18 9:27:00 AM |

[Figure]

Page 28: [34] Deleted          Joram Hooghiem          3/23/18 9:27:00 AM

[Figure]

| Page 28: [35] Formatted | Joram Hooghiem | 7/13/18 4:42:00 PM |
|---|---|---|

Font: 10 pt

| Page 28: [35] Formatted | Joram Hooghiem | 7/13/18 4:42:00 PM |
|---|---|---|

Font: 10 pt

| Page 28: [36] Formatted | Joram Hooghiem | 7/13/18 4:42:00 PM |
|---|---|---|

Font: 10 pt

| Page 28: [36] Formatted | Joram Hooghiem | 7/13/18 4:42:00 PM |
|---|---|---|

Font: 10 pt

| Page 28: [37] Deleted | Joram Hooghiem | 3/29/18 9:29:00 AM |
|---|---|---|

0

| Page 28: [37] Deleted | Joram Hooghiem | 3/29/18 9:29:00 AM |
|---|---|---|

0

| Page 28: [38] Deleted | Joram Hooghiem | 3/29/18 9:29:00 AM |
|---|---|---|

| Page 28: [38] Deleted | Joram Hooghiem | 3/29/18 9:29:00 AM |
|---|---|---|

| Page 28: [39] Deleted | Joram Hooghiem | 3/29/18 9:29:00 AM |
|---|---|---|

| Page 28: [39] Deleted | Joram Hooghiem | 3/29/18 9:29:00 AM |
|---|---|---|

| Page 30: [40] Change | Unknown | |
|---|---|---|

Field Code Changed

| Page 30: [40] Change | Unknown | |
|---|---|---|

Field Code Changed

| Page 30: [41] Deleted | Joram Hooghiem | 7/6/18 10:58:00 AM |
|---|---|---|

Table 6: Expected bias due to the limited accuracy of the LISA sampler. Typical values for the troposphere and stratosphere are taken from the indicated references: A) (Trolier et al., 1996) B) (Mrozek et al., 2016) C) (Nisbet et al., 2016) D) (Bergamaschi et al., 2001) E) (Aoki et al., 2003) and F) (Röckmann et al., 2011). $f_c$ was calculated using a source value 390 ppm ($CO_2$) and 1000 ppb ($CH_4$); contamination values of 0.55 ppm ($CO_2$) and 5.1 ppb ($CH_4$). Reported measurement reproducibility's for stratospheric air are also provided.

| Measurement Reproducibility (‰) |
|---|
| 0.02 (E) |
| 0.05 (E) |
| 0.2 (B) |
| 0.7 (F) |
| 2.3 (F) |

| Page 30: [42] Formatted | Joram Hooghiem | 7/13/18 4:42:00 PM |
|---|---|---|

Font: Times New Roman, Font color: Auto, English (UK)

| Page 30: [42] Formatted | Joram Hooghiem | 7/13/18 4:42:00 PM |
|---|---|---|

Font: Times New Roman, Font color: Auto, English (UK)

| Page 30: [42] Formatted | Joram Hooghiem | 7/13/18 4:42:00 PM |
|---|---|---|

Font: Times New Roman, Font color: Auto, English (UK)

| Page 30: [42] Formatted | Joram Hooghiem | 7/13/18 4:42:00 PM |
|---|---|---|

Font: Times New Roman, Font color: Auto, English (UK)

| Page 30: [42] Formatted | Joram Hooghiem | 7/13/18 4:42:00 PM |
|---|---|---|

Font: Times New Roman, Font color: Auto, English (UK)

| Page 30: [42] Formatted | Joram Hooghiem | 7/13/18 4:42:00 PM |
|---|---|---|

Font: Times New Roman, Font color: Auto, English (UK)

| Page 30: [42] Formatted | Joram Hooghiem | 7/13/18 4:42:00 PM |
|---|---|---|

Font: Times New Roman, Font color: Auto, English (UK)

| Page 30: [42] Formatted | Joram Hooghiem | 7/13/18 4:42:00 PM |
|---|---|---|

Font: Times New Roman, Font color: Auto, English (UK)

| Page 30: [42] Formatted | Joram Hooghiem | 7/13/18 4:42:00 PM |
|---|---|---|

Font: Times New Roman, Font color: Auto, English (UK)

| Page 30: [43] Formatted | Joram Hooghiem | 7/13/18 4:42:00 PM |
|---|---|---|

Font: 10 pt, English (UK)

| Page 30: [43] Formatted | Joram Hooghiem | 7/13/18 4:42:00 PM |

Font: 10 pt, English (UK)

| Page 30: [43] Formatted | Joram Hooghiem | 7/13/18 4:42:00 PM |

Font: 10 pt, English (UK)

| Page 30: [44] Formatted | Joram Hooghiem | 7/13/18 4:42:00 PM |

Font: 10 pt, English (UK)

| Page 30: [44] Formatted | Joram Hooghiem | 7/13/18 4:42:00 PM |

Font: 10 pt, English (UK)

| Page 30: [44] Formatted | Joram Hooghiem | 7/13/18 4:42:00 PM |

Font: 10 pt, English (UK)

| Page 30: [45] Formatted | Joram Hooghiem | 7/13/18 4:42:00 PM |

Font: 10 pt, English (UK)

| Page 30: [45] Formatted | Joram Hooghiem | 7/13/18 4:42:00 PM |

Font: 10 pt, English (UK)

| Page 30: [45] Formatted | Joram Hooghiem | 7/13/18 4:42:00 PM |

Font: 10 pt, English (UK)

| Page 30: [46] Formatted | Joram Hooghiem | 7/13/18 4:42:00 PM |

Font: 10 pt, English (UK)

| Page 30: [46] Formatted | Joram Hooghiem | 7/13/18 4:42:00 PM |

Font: 10 pt, English (UK)

| Page 30: [46] Formatted | Joram Hooghiem | 7/13/18 4:42:00 PM |

Font: 10 pt, English (UK)

| Page 30: [47] Formatted | Joram Hooghiem | 7/13/18 4:42:00 PM |

Font: 10 pt, English (UK)

| Page 30: [47] Formatted | Joram Hooghiem | 7/13/18 4:42:00 PM |

Font: 10 pt, English (UK)

| Page 30: [47] Formatted | Joram Hooghiem | 7/13/18 4:42:00 PM |

Font: 10 pt, English (UK)

| Page 30: [48] Formatted | Joram Hooghiem | 7/13/18 4:42:00 PM |

Font: 10 pt, English (UK)

| Page 30: [48] Formatted | Joram Hooghiem | 7/13/18 4:42:00 PM |

Font: 10 pt, English (UK)

| Page 30: [48] Formatted | Joram Hooghiem | 7/13/18 4:42:00 PM |

Font: 10 pt, English (UK)

| Page 30: [49] Formatted | Joram Hooghiem | 7/13/18 4:42:00 PM |

Font: 10 pt, English (UK)

| Page 30: [49] Formatted | Joram Hooghiem | 7/13/18 4:42:00 PM |

Font: 10 pt, English (UK)

| Page 30: [49] Formatted | Joram Hooghiem | 7/13/18 4:42:00 PM |
|---|---|---|

Font: 10 pt, English (UK)

| Page 30: [50] Formatted | Joram Hooghiem | 7/13/18 4:42:00 PM |
|---|---|---|

Font: 10 pt, English (UK)

| Page 30: [50] Formatted | Joram Hooghiem | 7/13/18 4:42:00 PM |
|---|---|---|

Font: 10 pt, English (UK)

| Page 30: [50] Formatted | Joram Hooghiem | 7/13/18 4:42:00 PM |
|---|---|---|

Font: 10 pt, English (UK)

| Page 30: [51] Formatted | Joram Hooghiem | 7/13/18 4:42:00 PM |
|---|---|---|

Font: 10 pt, English (UK)

| Page 30: [51] Formatted | Joram Hooghiem | 7/13/18 4:42:00 PM |
|---|---|---|

Font: 10 pt, English (UK)

| Page 30: [51] Formatted | Joram Hooghiem | 7/13/18 4:42:00 PM |
|---|---|---|

Font: 10 pt, English (UK)

| Page 30: [52] Formatted | Joram Hooghiem | 7/13/18 4:42:00 PM |
|---|---|---|

Font: 10 pt, English (UK)

| Page 30: [52] Formatted | Joram Hooghiem | 7/13/18 4:42:00 PM |
|---|---|---|

Font: 10 pt, English (UK)

| Page 30: [52] Formatted | Joram Hooghiem | 7/13/18 4:42:00 PM |
|---|---|---|

Font: 10 pt, English (UK)

| Page 30: [53] Formatted | Joram Hooghiem | 7/13/18 4:42:00 PM |
|---|---|---|

Font: 10 pt, English (UK)

| Page 30: [53] Formatted | Joram Hooghiem | 7/13/18 4:42:00 PM |
|---|---|---|

Font: 10 pt, English (UK)

| Page 30: [53] Formatted | Joram Hooghiem | 7/13/18 4:42:00 PM |
|---|---|---|

Font: 10 pt, English (UK)

| Page 30: [54] Change | Unknown | |
|---|---|---|

Field Code Changed

| Page 30: [54] Change | Unknown | |
|---|---|---|

Field Code Changed

| Page 30: [54] Change | Unknown | |
|---|---|---|

Field Code Changed

| Page 30: [54] Change | Unknown | |
|---|---|---|

Field Code Changed

| Page 30: [54] Change | Unknown | |
|---|---|---|

Field Code Changed

| Page 30: [54] Change | Unknown | |
|---|---|---|

Field Code Changed

---

## Author Response (AR2)

The authors have answered most of the issues brought forward during the review process. The calculation of the contamination in section 6.4., however, still seems to be odd and in my view is flawed.

The argumentation before eq. 6 is clearly wrong. The majority of the molecules in the sample bag are not introduced via diffusion (you would need a very long sampling time) but by whole air flux.

Second, I cannot follow the argumentation that all the contamination is explained by diffusion, either. Small leaks in the system would also allow for introduction of a contamination, which would not be driven by diffusion.

Third, I cannot follow the mathematical argumentation. If a diffusive flux into the sample bag or out of the bag is calculated (eq. 8 and 9) a concentration gradient is required (not a concentration). Eq. 8 and 9 are in my view not correct and also do not yield fluxes, simply when looking at the units. Then in eq. 9 the differences between the two fluxes is used to derive a bias in mixing ratio. This is again physically not correct (the difference of two fluxes would be a flux). Mixing ratios and fluxes are again mixed up in eq. 10-12.

Then again, the step from (15) to (16) does not seem to involve the usage of (8) and (9) as stated but rather is based on (13) and (14).

In my view, equation 16 is correct, but does not require any argumentation about molecular diffusion, but can simply be derived from 13 and 14. In fact, if it is to be derived based on assumptions about molecular diffusion, then the diffusion coefficients of the different isotopes would need to be considered. The fraction fc which is then needed in (16) can simply be estimated as explained in my first review from mass balance argumentation. This would then yield very similar results as found in the revised manuscript. The way it is derived in the revised manuscript is, however, in my view, not correct. I think that arguing just by assuming that it must be a contamination from mixing with air of different origin and thus applying mass balance equations is much simpler and sufficient for such an estimation.

We agree with the reviewer that the derivation in section 6.4 is lacking a thorough analysis. We were aware of the fact that small leaks in the system would allow for introduction of contamination air. The reason why we tried to derive the fraction of the contamination air entirely based on diffusion is that this would systematically explain the observed biases or part of the biases, rather than somewhat random if it would be due to leaks.

Furthermore, the reviewer is right to point out that the results found in the revised manuscript is similar to the reviewer's original calculation. We therefore have adopted the reviewer's simpler mass balance for the estimation, as it would communicate the same message.

[revised manuscript text omitted]